# Feasible supply of steel and cement within a carbon budget is likely to fall short of expected global demand

Takuma Watari [1,2] ✉, André Cabrera Serrenho [2], Lukas Gast [2], Jonathan Cullen [2] & Julian Allwood [2]

The current decarbonization strategy for the steel and cement industries is inherently dependent on the build-out of infrastructure, including for $CO_2$ transport and storage, renewable electricity, and green hydrogen. However, the deployment of this infrastructure entails considerable uncertainty. Here we explore the global feasible supply of steel and cement within Paris-compliant carbon budgets, explicitly considering uncertainties in the deployment of infrastructure. Our scenario analysis reveals that despite substantial growth in recycling- and hydrogen-based production, the feasible steel supply will only meet 58–65% (interquartile range) of the expected baseline demand in 2050. Cement supply is even more uncertain due to limited mitigation options, meeting only 22–56% (interquartile range) of the expected baseline demand in 2050. These findings pose a two-fold challenge for decarbonizing the steel and cement industries: on the one hand, governments need to expand essential infrastructure rapidly; on the other hand, industries need to prepare for the risk of deployment failures, rather than solely waiting for large-scale infrastructure to emerge. Our feasible supply scenarios provide compelling evidence of the urgency of demand-side actions and establish benchmarks for the required level of resource efficiency.

Steel and cement are staples of our daily lives. The cars we drive, the buildings we inhabit, and the infrastructure that allows us to travel from place to place are all supported by abundant and cheap materials. But that abundance and cheapness come at a price: carbon emissions that cause devastating climate change[1]. Global steel and cement production has more than doubled over the past 20 years and now accounts for about 15% of global $CO_2$ emissions[2]. Clearly, the way we produce and use steel and cement needs to be transformed to achieve zero emissions by around the middle of the 21st century, an absolute requirement for a stable climate[3]. But how is it possible to decarbonize steel and cement production within such a limited timeframe while still providing essential services to a growing world population?

Existing decarbonization scenarios have usually emphasized the rapid and large-scale deployment of supply-side technologies,

including carbon capture and hydrogen technologies[4]. For example, a series of Energy Technology Perspectives reports by the International Energy Agency (IEA) expects that, on average, 40% and 60% of 2050 emission reductions for the steel and cement sectors, respectively, will come from carbon, capture utilization, and storage (CCUS) (Supplementary Fig. 1). This trend is also reflected in the academic literature on steel[5], cement[6], or the entire industrial sector[7], where a significant part of the emission reductions is expected to come from CCUS. An emerging body of literature focuses more on hydrogen technologies, particularly direct reduced iron (DRI) using green hydrogen[8]. Recent evidence indicates that hydrogen-based steel can be economically competitive when combined with high-quality iron ore, low steelworker wages, and abundant and cost-effective renewable electricity[9]. However, both CCUS and hydrogen-based solutions share a common

[1]Material Cycles Division, National Institute for Environmental Studies, Tsukuba, Japan. [2]Department of Engineering, University of Cambridge, Cambridge, UK. ✉e-mail: watari.takuma@nies.go.jp

challenge: the indispensability of infrastructure. These decarbonization strategies essentially rely on large-scale build-out of infrastructure, including for $CO_2$ transport and storage, renewable electricity, and green hydrogen. In particular, steel decarbonization benefits from all of these technologies, while cement decarbonization relies heavily on CCUS[10]. This perspective raises the question of the feasibility of large-scale and rapid deployment of infrastructure.

Addressing this question fully is undoubtedly a complex task having economic, social, and environmental aspects[11], but the historical data reveal two key considerations. The first is the preparation period. CCUS and green hydrogen production are mature technologies with a high level of technology readiness[12], but they have not yet been deployed at scale. The implementation of these technologies requires a period of preparation, involving pilot studies at increasing scale, connection to existing infrastructure, legal permissions, social consent, and financial acquisition, all before the actual deployment period. Based on the history of energy infrastructure, this set of processes could take decades[13]. The second is the rate of deployment. One pioneering study demonstrates that even if starting today electrolysis capacity grows as fast as wind and solar power have done, the green hydrogen supply is unlikely to reach the desired levels[14]. Infrastructure deployment inevitably takes a long time, due to the realities of engineering and construction, including the political and regulatory procedures[15].

Collectively, the steel and cement industries face a critical risk of uncertain infrastructure deployment over which they have no direct control. Nevertheless, the existing literature implicitly presumes large-scale infrastructure build-out[16], resulting in an under-emphasis of the risk of deployment failure. This study aims to fill this gap by calculating global steel and cement supply in line with Paris-compliant carbon budgets, explicitly considering uncertainties in the build rate of

infrastructure. Specifically, we consider two types of zero-emissions infrastructure: CCUS and non-emitting electricity. The analysis is performed using a stochastic optimization model based on physical mass balancing, rather than complex, large-scale economic models. While this is a simplified, aggregated model, it allows future infrastructure deployment to be explicitly linked to the decarbonization of steel and cement production with transparent assumptions.

## Results

### Uncertain infrastructure deployment

The analysis begins by exploring the potential range of future deployment of infrastructure by understanding the historical build-out and future scenarios based on the IEA's database[17]. A deep dive into the zero-emissions infrastructure-related data reveals two key insights.

First, current CCUS capacity falls short of the levels that past IEA reports assumed would be deployed by 2021 (Fig. 1a). For instance, the 2010 IEA report assumed that $CO_2$ capture for the steel and cement sectors would reach ~195 million metric tons (Mt)-$CO_2$ in 2021 but the current operating capacity for the steel and cement sectors is just under 1 Mt-$CO_2$[18]. It appears that CCUS-related infrastructure has not been deployed as originally planned. Second, the 2050 CCUS capacity envisaged in the IEA scenarios requires an expansion at a rate that far exceeds current construction plans (Fig. 1b). Despite the historical failure of CCUS deployment, the IEA scenarios consistently assume ~2000 Mt-$CO_2$ capture in the steel and cement sectors for 2050, which is 2000 times the current capacity for these sectors (~1 Mt-$CO_2$) and more than 100 times the 2030 construction plan (~19 Mt-$CO_2$)[18].

This is not to say that the IEA scenarios are physically or economically unrealistic, but that their realization is highly uncertain given the scale of the challenge and our historical failures. Therefore, this study makes the following assumptions about CCUS deployment

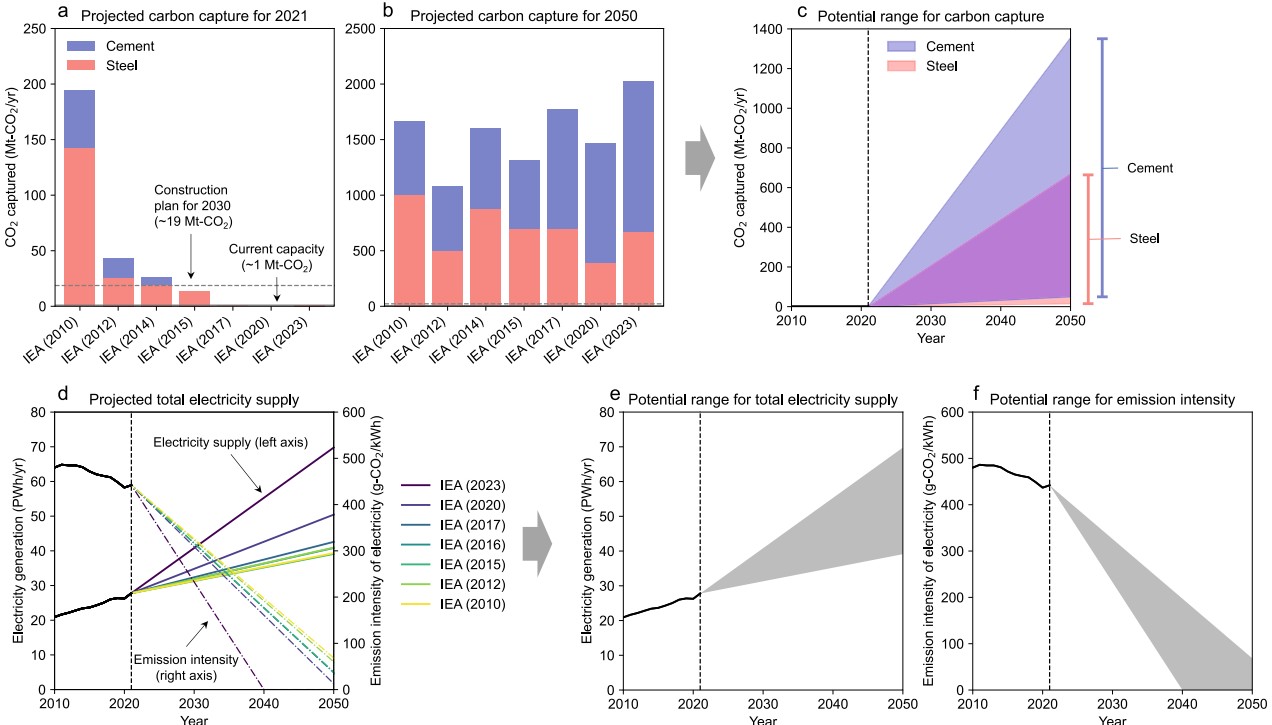

**Fig. 1 | Potential range of future global zero-emissions infrastructure deployment. a** Carbon capture capacity for 2021 projected by the International Energy Agency (IEA) scenarios. **b** Carbon capture capacity for 2050 projected by the IEA scenarios. **c** The potential range of future carbon capture capacity. **d** Total electricity supply projected by the IEA scenarios. **e** The potential range of future total electricity supply. **f** The potential range of future emission intensity of electricity.

The data are based on a series of Energy Technology Perspectives reports[17]. We examined all reports and extracted data from those reports for which data were available. Current operating and planned carbon capture capacities for 2030 were obtained by accessing the IEA database in June 2023[18]. The right-hand error bars in Fig. 1c show the range of 2050 values.

(Fig. 1c): First, the upper bound of CCUS deployment is based on the 2023 IEA report[19], which considers the most ambitious CCUS capacity in the steel and cement sectors combined. This scenario envisions the steel and cement sectors achieving CCUS capacities of 670 Mt-$CO_2$ and 1355 Mt-$CO_2$, respectively, by 2050. Second, the lower bound is derived from a more conservative linear extrapolation based on the current operating capacity and the 2030 construction plan. This conservative estimate yields CCUS capacities of 15 Mt-$CO_2$ and 50 Mt-$CO_2$ for the steel and cement sectors, respectively, by 2050.

A similar approach is taken for non-emitting electricity supply. The IEA scenarios tend to assume higher levels of total non-emitting electricity supply in more recent reports (Fig. 1d). This may reflect two factors: the success of cost reductions in solar and wind, and the failure of emission reductions, leading to more stringent electrification requirements now and in the future[20]. However, despite the substantial expansion of non-emitting electricity supply, it still falls short of the annual increase needed to meet the most ambitious scenario[21]. This study assumes that the scenario ranges reflect the uncertainty associated with the future deployment of non-emitting electricity. Therefore, the upper and lower bounds for both electricity supply and its emission intensity are established based on the IEA scenario ranges (Fig. 1e, f).

Given the increasing demand for non-emitting electricity across various sectors due to electrification[22], this study assumes that the electricity available for materials production can increase proportionally to the total electricity supply. Our analysis specifically focuses on electricity supply as a relevant indicator of green hydrogen uptake, although various infrastructure components (i.e., electrolyzers, storage tanks, and transport facilities) also need to be built. This

assumption is grounded in the understanding that affordable and reliable electricity is a fundamental requirement for the competitiveness of green hydrogen-based DRI[9].

The potential ranges of zero-emissions infrastructure deployment, as depicted in Fig. 1c, e, f, are fed into the optimization model. The model attempts to maximize the global supply of steel and cement under the constraints of carbon budgets, infrastructure deployment, and scrap availability through physical mass balancing. To avoid arbitrary judgments about the most likely value of zero-emissions infrastructure deployment or the shape of its distribution, a uniform distribution is used for each variable (i.e., Fig. 1c, e, f) between its upper and lower bounds[23]. The carbon budgets employed in the model are based on limiting the global mean temperature rise within 1.5 °C with a 50% probability, consistent with the Paris Agreement[3]. We also consider a carbon budget for a 50% probability of 1.7 °C (equivalent to an 83% probability of 2.0 °C), which corresponds to a well-below 2 °C budget[24]. To isolate the impact of infrastructure uncertainty on the supply of steel and cement, we presume that technological progress in these industries will align with established roadmaps and industry-accepted best practices (see Methods section).

## Limited feasible materials supply within carbon budgets

We now estimate the global feasible supply of steel and cement within Paris-compliant carbon budgets, with explicit consideration of the uncertain deployment of zero-emissions infrastructure. Figure 2a, b show that despite significant technological advances within the materials industry, the feasible supply of steel and cement within the 1.5 °C budget is likely to fall short of the expected demand. The model estimates that by 2050, the feasible steel supply could be only 75–84%

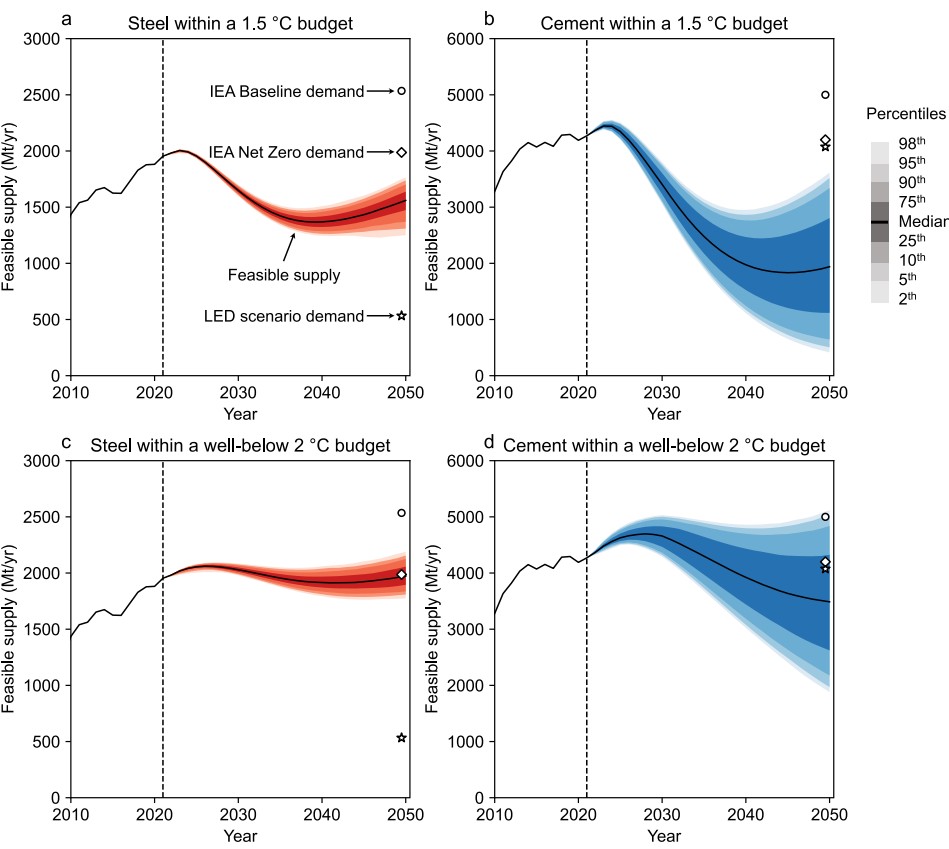

**Fig. 2 | Global feasible supply of steel and cement within Paris-compliant carbon budgets by 2050. a** Steel supply within a 1.5 °C budget. **b** Cement supply within a 1.5 °C budget. **c** Steel supply within a well-below 2 °C budget. **d** Cement supply within a well-below 2 °C budget. The expected demand data are based on the International Energy Agency (IEA) Baseline scenario (Stated Policies Scenario)[25], the IEA Net Zero scenario[26], and the Low Energy Demand (LED) scenario[27].

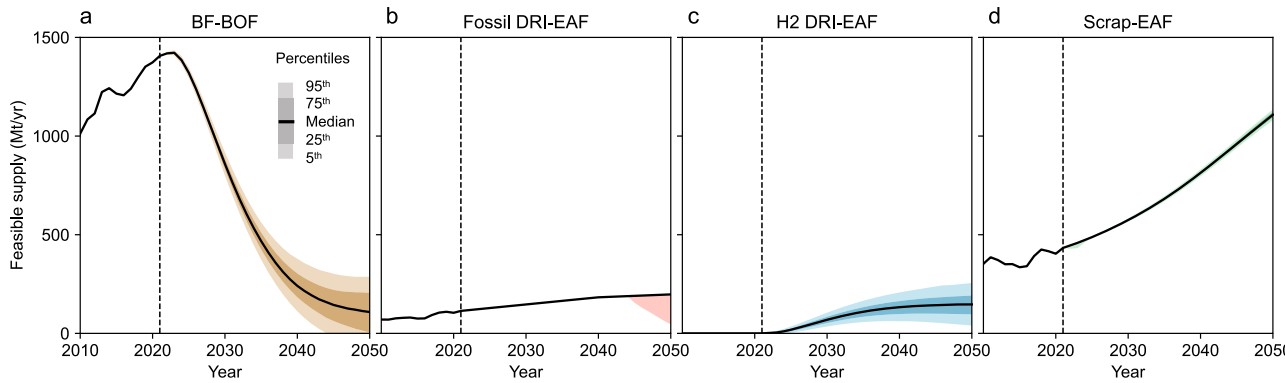

**Fig. 3 | Global crude steel production per process within a 1.5 °C budget by 2050. a** BF-BOF: blast furnace and basic oxygen furnace route. **b** Fossil DRI-EAF: fossil fuel-based direct reduced iron and electric arc furnace route. **c** H2 DRI-EAF: hydrogen-based direct reduced iron and electric arc furnace route. **d** Scrap-EAF: scrap-based electric arc furnace route.

(interquartile range) of the current total supply. Cement supply could become even more scarce due to limited mitigation options, remaining at 26–66% (interquartile range) of current levels by 2050. These trends suggest that continued growth in production is unlikely in the face of the limited carbon budget unless we build out essential infrastructure at an unprecedented scale within a limited timeframe.

Comparing these feasible supply levels to the expected demands reveals a significant mismatch. The feasible supply levels correspond to 58–65% (interquartile range) for steel and 22–56% (interquartile range) for cement, relative to the 2050 expected baseline demands[25]. Even if modest demand reduction strategies are materialized, as outlined in the IEA Net Zero scenario[26], the level of feasible supply does not match the expected demand. The only major existing scenario in which the expected demand for steel could be fully met by the feasible supply is the Low Energy Demand (LED) scenario[27], which includes the most ambitious demand reduction measures. However, even in this scenario, demand for cement is not fully met, suggesting a more severe supply-demand gap. These trends hold true even with the well-below 2 °C budget (Fig. 2c, d); the feasible supply is likely to fall short of the expected baseline demand, although the modest demand reduction scenarios fall within the interquartile range of the feasible supply.

A comparison of steel and cement highlights three caveats. First, the uncertainty in cement supply is greater than the uncertainty in steel supply. This is due to the relatively limited options for decarbonizing the cement production process, which depends exclusively on the deployment of CCUS. Second, the range of expected demand is greater for steel than for cement. This discrepancy likely stems from the limited exploration of demand reduction strategies for cement. In fact, the LED scenario[27] only considers one strategy for cement due to the lack of scientific evidence, while a broader range of strategies are examined for steel. Third, the feasible supply of both steel and cement exhibits an initial downward trend, followed by an upward trend. This pattern is attributed to the combined effects of the shape of the carbon budget, which requires immediate emission reductions, and the linear deployment of zero-emission infrastructure, which is assumed to increase steadily over time.

The results presented in this section are not predictions of what will actually happen in the future. Rather, they serve as a warning of the risks associated with relying solely on infrastructure deployment to decarbonize the steel and cement industries. If we leave industrial decarbonization entirely to infrastructure deployment, we may fail to prepare for a significant future shortfall between feasible supply and expected demand.

## Certain growth of steel recycling in an uncertain future
The relatively modest shortfall and uncertainty of feasible steel supply are due to the diversity of production processes, as shown in Fig. 3.

While blast furnace-based production will inevitably be scaled down to align with the 1.5 °C budget, hydrogen-based and recycling-based production, powered by non-emitting electricity, both have significant growth opportunities. The model estimates that these two production routes could provide the equivalent of more than 60% of the current total supply in 2050, with interquartile ranges of 100-200 Mt and 1100-1120 Mt in 2050, respectively. Due to lower electricity use per unit of production, the recycling-based route shows a more robust growth than the hydrogen-based route, but is limited by scrap availability. Fossil fuel-based DRI production, while having a better emission profile than blast furnace-based production, is highly dependent on regional fossil fuel availability[28]. Consequently, its production levels are expected to experience moderate changes (Supplementary Fig. 5). These trends remain the same with the well-below 2 °C budget; the difference between the 1.5 °C budget and the well-below 2 °C budget is mainly in the rate of phase-out of blast furnace-based production, with the well-below 2 °C budget showing a more gradual phase-out by 2050 (Supplementary Fig. 7).

Again, the results presented here are not predictions of the future. They indicate the varying degrees of vulnerability of each production route to the uncertain infrastructure deployment. The ore-based production is more sensitive to the level of infrastructure deployment, while recycling-based production remains relatively unscathed, so growth is more certain.

## Inequality as a major challenge to meeting basic human needs
Given the potential supply shortfall, an emerging question is whether it is possible to satisfy the basic needs of a growing world population with the feasible supply of steel and cement. To answer this question, we extract data on the minimum material requirements for satisfying basic human needs from an empirical study analyzing the relationship between global in-use steel and cement stocks and the five essential services (i.e., electricity, water, sanitation, shelter, and mobility)[29].

A comparison of supply and demand shows that both steel and cement have a feasible supply that could fully meet the basic needs of the growing world population, even within the 1.5 °C budget (Fig. 4). Cumulatively, the minimum requirements to meet basic human needs by 2050 are only 13–14% (interquartile range) of the feasible supply for steel and 52–63% (interquartile range) for cement in the case of the 1.5 °C budget. The minimum requirements represent an even smaller fraction of the feasible supply under the well-below 2 °C budget: 11–12% (interquartile range) for steel and 39–45% (interquartile range) for cement. Theoretically, therefore, the basic needs of the growing world population could well be met by the feasible supply, even if the zero-emissions infrastructure is deployed at the lower end of the potential range.

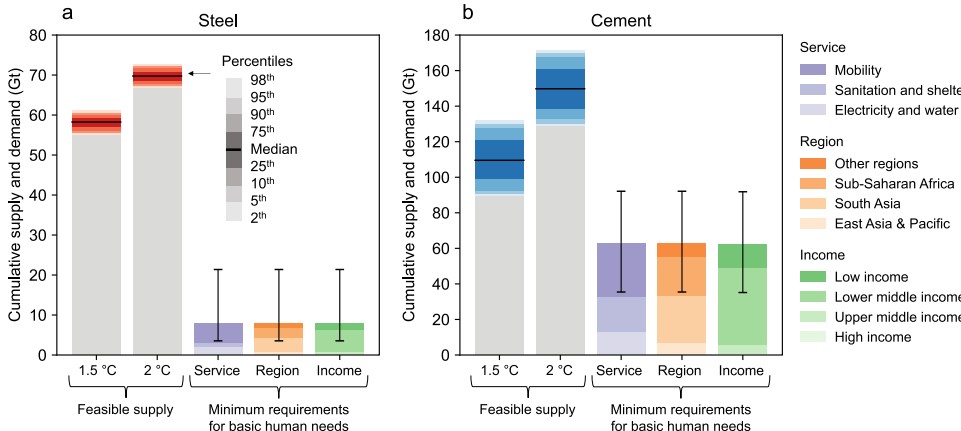

**Fig. 4 | Cumulative feasible supply of steel and cement compared to the minimum requirements to meet basic human needs, 2015-2050. a** Steel. **b** Cement. The data on the minimum requirements are based on empirical data on the relationship between global in-use steel and cement stocks and the five essential services (i.e., electricity, water, sanitation, shelter, and mobility)[29]. Error bars reflect the uncertainty in the relationship between historical need satisfaction and material use levels around the world. It should be noted that the minimum material requirements for basic human needs only include the requirements for countries where the per capita in-use material stocks do not reach sufficient levels to satisfy needs[29].

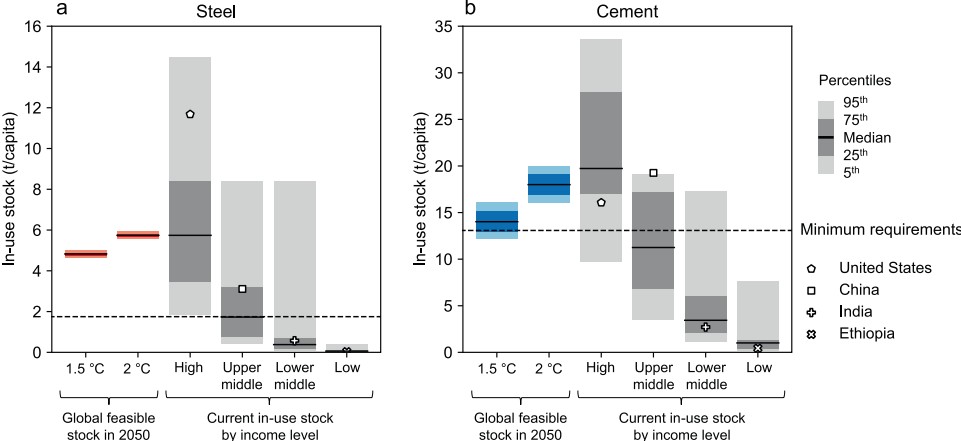

**Fig. 5 | Feasible per capita in-use stocks of steel and cement in 2050 compared to the current levels across four income groups. a** Steel. **b** Cement. Current in-use stock levels for steel and cement are based on the literature[29] and divided into four income groups with reference to the World Bank classification. The most populous countries in each income group (i.e., the United States, China, India, and Ethiopia) are presented separately from the overall distribution. The levels of minimum requirements to satisfy basic human needs are based on the literature[29].

However, fundamental challenges are posed by the ever-growing needs of high-income countries and the 'equitable' distribution of the feasible supply of steel and cement. As shown in Fig. 5, current per capita material stocks in high-income countries far exceed the feasible stock levels derived from the 1.5–2 °C budgets, while low-income countries have material stocks well below the feasible levels. However, it is possible to meet basic human needs across the world population with the feasible stock levels. The challenge lies in the substantial material demand generated by high-income countries for stock replacement and expansion, with houses and cars becoming larger and heavier year after year[30]. These trends suggest that inequalities in the use of steel and cement, if not addressed head-on, could impede the provision of the basic needs of the global population within the feasible supply. Since more than 90% of material requirements to meet basic human needs will come from lower-middle and low-income countries, predominantly in South Asia and sub-Saharan Africa, the primary challenge will be to balance the ever-growing demands of high-income countries[31], with the need to distribute the feasible supply equitably.

## Discussion

The current decarbonization strategy for the steel and cement industries is inherently dependent on the deployment of infrastructure, which is highly uncertain in the face of technical, economic, and social challenges. Our analysis demonstrates the risk of simply waiting for the infrastructure to emerge; despite significant technological advances within these industries, the feasible supply of steel and cement in line with Paris-compliant carbon budgets is likely to fall short of expected demand.

This finding poses a twofold challenge for decarbonizing the steel and cement industries: on the one hand, governments need to expand essential infrastructure rapidly; on the other hand, industries need to be well prepared for the risk of deployment failures. For the materials industry, this could involve strategic investments in decarbonized production processes, as well as adding more value to materials by selling them as services rather than as commodities[32]. For the construction and manufacturing industries, this could mean providing the same level of services with less material by changing the way products are designed, used, and disposed of[33].

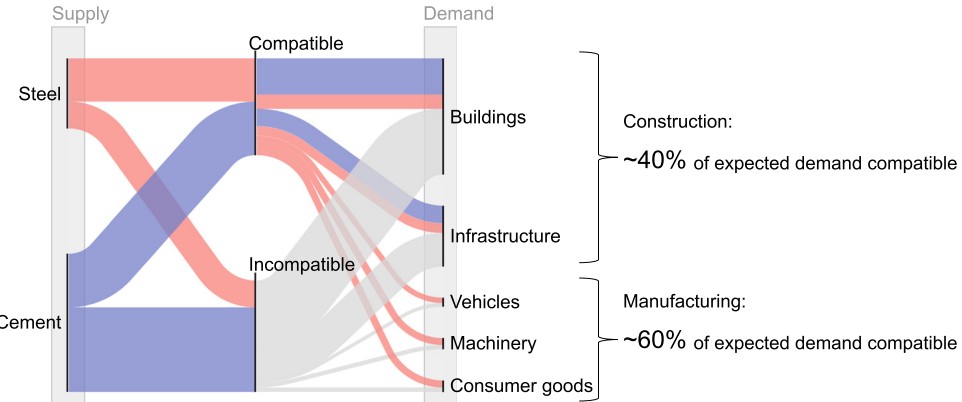

**Fig. 6 | Comparison of the global feasible supply and expected demand for the construction and manufacturing sectors in 2050.** The feasible supply is based on the 1.5 °C budget case. The expected demand is based on the International Energy Agency Baseline scenario (Stated Policies Scenario)[25]. 'Compatible' indicates feasible supply, while 'Incompatible' indicates a gap between feasible supply and expected demand.

To get a sense of the challenge, we allocate the feasible supply to each end-use application based on the current share[25] (Fig. 6). The feasible supply of steel and cement consistent with the 1.5 °C budget may only meet ~40% of the expected baseline demand for construction and ~60% for manufacturing, based on our median estimate. This perspective is critical because the construction and manufacturing industries are largely unaware that the feasible supply is likely to fall short of the expected demand, and are thus under-prepared for that future[34].

Our analysis provides clear signals for the related stakeholders to promote a suite of demand-side resource efficiency strategies[35]. For example, there is an important opportunity to almost halve the use of materials in the building frame through optimized design and reduced overdesign[36]. Strategic urban planning that encourages the shift from private cars to public transport can reduce the material use per distance traveled[37]. Upfront design for material utilization and flexible blanking equipment can significantly reduce material losses in manufacturing[38]. Material loss during construction can also be reduced by improving the architectural and engineering specifications and addressing a lack of client interest[39]. Products can last longer through changing consumer behaviors[40]. There is also evidence that a large portion of end-of-life material components can be reused without melting down if we have the will to do so[41]. Existing evidence suggests that the construction and manufacturing industries can provide the same level of services with around 35–70% and 55–65% less material use, respectively (Supplementary Tables 5 and 6). Such behavioral and cultural changes can happen much faster than technological change but are often left out of decarbonization plans[42]. What is needed, therefore, is a holistic and balanced discussion to encourage actions in these domains, rather than solely waiting for large-scale infrastructure to emerge.

In this context, we argue for the importance of a 'forecasting supply and backcasting demand' modeling approach, rather than the traditional 'forecasting demand and backcasting supply'. To date, most modeling studies have first forecasted material and energy demand and subsequently backcasted structures of supply-side technology to meet that demand[43]. This approach inevitably leads to a heavy reliance on accelerated infrastructure deployment and focuses less on demand-side actions[44]. Instead, as demonstrated in this study, we can highlight the urgent necessity of demand-side actions by first better understanding the feasible supply and then exploring how to use it on the demand side. The proposed approach provides clearer evidence and incentives for related stakeholders to take action by demonstrating the limited material availability. The approach is also advantageous because it indicates the specific level of resource efficiency required under the feasible supply: we will need to provide the same level of services with 60% less material use in construction and 40% less in manufacturing to stay within the 1.5 °C budget. While previous studies have also demonstrated the importance of demand-side actions in decarbonizing the production of steel[45] and cement[46], this study, based on the 'forecasting supply and backcasting demand' approach, puts this importance in a more vivid context; the feasible supply of steel and cement is likely to fall short of desired levels unless we build out essential infrastructure at an unprecedented scale in a limited timeframe.

We are not saying that our scenario is the only feasible pathway to decarbonize the steel and cement industries or more feasible than the other scenarios. Rather, the results highlight the risk of simply waiting for infrastructure to emerge. Based on the precautionary principle, the materials and related industries should prepare for an uncertain future through proactive measures, while at the same time stimulating governments to deploy essential infrastructure rapidly. The key challenge for governments and international organizations will then be how to distribute the feasible supply on an equitable basis to meet the basic needs of the growing world population. Greater responsibility should be placed on high-income countries, which have much larger in-use material stocks than low-income countries and can enjoy a 'scrap privilege' from their past production and carbon emissions[47]. If these challenges are not addressed head-on, the gap between feasible supply and expected demand risks being filled by emission-intensive production processes. This could result in cumulative emissions of up to ~160 Gt-$CO_2$ by 2050, representing ~40% of the remaining 1.5 °C budget or ~20% of the remaining well-below 2 °C budget (Supplementary Fig. 9). Avoiding this future depends on how the current generation prepares for an uncertain future; we must not leave it to the enormous efforts of future generations.

## Methods

### Model overview

We construct an optimization model based on physical mass balancing equations to explore feasible materials supply in a carbon-constrained world. The model estimates the maximum global materials production within Paris-compliant carbon budgets, given the potential range of zero-emissions infrastructure deployment: CCUS and non-emitting electricity. The deployment of each production process is selected endogenously according to its profile regarding carbon emissions, electricity use, and resource requirements. The core equations of the model for the case of steel consist of Eqs. (1) to (8) below. In the case of cement, the model consists of Eqs. (1) to (3), as there is limited variation in the production process.

Maximizing:

$$\sum_t \sum_i P_i(t) \tag{1}$$

Subject to:

$$\sum_i P_i(t)\left(CI_i^{(f)}(t) + EI_i(t)CI^{(e)}(t)\right) + CC(t)\left(EI_{cc}CI^{(e)}(t) - 1\right) \le Cap_{Carbon}(t) \tag{2}$$

$$\sum_i P_i(t)EI_i(t) + CC(t)EI_{cc} \le Cap_{Electricity}(t) \tag{3}$$

$$P_{F\_DRI}(t) \le Cap_{F\_DRI}(t) \tag{4}$$

$$P_{EAF}(t) \le Cap_{EAF}(t) \tag{5}$$

where:

$$Cap_{EAF}(t) = \theta\big(Pre_{EAF}(t) + Post(t)\big) \tag{6}$$

$$Pre_{EAF}(t) = (1-\delta)(1-\lambda)\sum_i P_i(t) - \omega P_{BOF}(t) \tag{7}$$

$$Post(t) = \gamma(t)(1-\rho)\sum_{t'=0}^{t}\left(\delta\lambda\sum_i P_i(t)\,\phi(t-t')\right) \tag{8}$$

in which system variables and parameters are defined as follows. $P_i(t)$: Material production in route $i$. $CI_i^{(f)}(t)$: Emission intensity of material production in route $i$ due to fuel combustion and chemical process. $EI_i(t)$: Electricity intensity of material production in route $i$. $CI^{(e)}(t)$: Emission intensity of electricity use. $CC(t)$: Captured carbon through CCUS. $EI_{cc}$: Energy penalty of carbon capture technologies. $Cap_{Carbon}(t)$: Carbon budget. $Cap_{Electricity}(t)$: Maximum electricity supply. $Cap_{F\_DRI}(t)$: Maximum production via fossil DRI-EAF route. $Cap_{EAF}(t)$: Maximum production via scrap-EAF route. $Pre_{EAF}(t)$: Recovered pre-consumer scrap for use in scrap-EAF route. $Post(t)$: Recovered post-consumer scrap. $\theta$: EAF production yield. $\delta$: Forming yield. $\lambda$: Fabrication yield. $\omega$: Ratio of scrap in total BOF input. $\gamma(t)$: Recovery rate of post-consumer scrap. $\rho$: Hibernation ratio of end-of-life products. $\phi(t-t')$: Lifetime distribution.

The in-use material stocks are then estimated using a time-cohort-type approach, where the in-use stock is estimated each year based on the total inflow of materials embedded in the remaining products. Assuming that the flow of material into the in-use stock phase in year $t$ is $I(t)$ and the flow of material out of the in-use stock phase in year $t$ is $O(t)$, the in-use stock $S(t)$ can be calculated by simple mass balance:

$$S(t) = \sum_{t'=0}^{t}(I(t') - O(t')) \tag{9}$$

where:

$$O(t) = \sum_{t'=0}^{t} I(t')\,\phi(t-t') \tag{10}$$

The in-use material stocks are interpreted here as an approximate indicator of material services, since our demand for services is not met by the produced materials themselves, but by the in-use stocks accumulated in the form of products and infrastructure[48].

The model performs a Monte Carlo simulation, where $CC(t)$, $Cap_{Electricity}(t)$, and $CI^{(e)}(t)$ are randomly selected from the potential range (Fig. 1c, e, and f), and optimization is run 1000 times to derive the uncertainty range for the results. To avoid arbitrary judgments about the most likely value of infrastructure deployment or the shape of its distribution, a uniform distribution is used for each variable between its upper and lower bounds[23].

The main data sources are based on various academic papers and reports: system variables and parameters governing physical mass balance[49–52]; emission profiles of material production[5,53–58]; energy penalty of carbon capture[59,60]; and future population[61]. More details can be found in the Supplementary Tables 1, 2, 3, and 4.

## Carbon budgets
The carbon budget is based on limiting the global mean temperature rise within 1.5 °C with a 50% probability, consistent with the Paris Agreement pledges (~420 Gt-CO$_2$)[3]. We also consider a carbon budget for a 50% probability of 1.7 °C (equivalent to an 83% probability of 2.0 °C), which corresponds to "well below 2 °C" (~770 Gt-CO$_2$)[24]. We allocate the total carbon budgets to the global steel and cement sectors by multiplying the current emissions of the steel and cement sectors by the annual emissions mitigation rate[62] (Supplementary Figs. 3 and 4). This reflects the assumption that the global steel and cement sectors contribute to mitigation paths in proportion to other sectors, as has been assumed in several previous studies for steel[55], cement[63], and paper[64]. It is important to note here that the carbon budgets assumed here do not envisage large-scale carbon dioxide removal and therefore require more rapid emission reductions than those in the IEA Net Zero scenario[26].

## Technological progress
This study assumes that technological progress in the industry will align with established roadmaps and industry-accepted best practices. Specifically, we envision top gas recycling and coke substitution in blast furnaces and improved post-consumer scrap recovery for the steel sector. For the cement sector, our assumptions include improved energy efficiency, clinker substitution, and fuel substitution. Detailed information on the level of improvement can be found in the supplementary information.

In this domain, four key assumptions deserve attention. First, the hydrogen-based direct reduction system is based on the literature[56] with an electrolyzer efficiency of 45 kWh/kg H$_2$ and a hydrogen mass flow rate of 1.5 (i.e., 50% oversupply of hydrogen for full conversion of iron ore in the shaft[8]). Since this electrolyzer efficiency is already at a high level[65], no further efficiency improvement is assumed. Second, clinker has traditionally been substituted mainly by fly ash and granulated blast furnace slag (GBFS), which will decline in a decarbonized future[66]. Therefore, clinker substitution will have to be provided by resources other than fly ash and GBFS. Our assumption here is based on recent studies that show significant potential for substitution with calcined clay, agricultural by-product ash, forestry by-product ash, and end-of-life binders[67,68]. Third, similar to supplementary cementitious materials, the cement industry currently uses waste materials as thermal fuels, which may become less available in a decarbonized, more circular future[54]. We assume an expanded use of waste from agricultural, chemical, and food production[58]. It is important to note that a more comprehensive assessment of resource availability needs to consider the interconnected system beyond just the steel and cement industries[69]. Fourth, technological improvements are assumed to be achieved linearly over the period 2050. This assumption, although simplistic, is based on the limited evidence and is consistent with industry roadmaps that occasionally assume linear progress[58].

## Expected future demand

The estimated feasible supply is compared with the expected demand in three major existing scenarios: the IEA Baseline scenario, in which demand grows without intervention[25]; the IEA Net Zero scenario, in which modest demand reduction strategies are materialized[26]; and the Low Energy Demand scenario, in which the most ambitious demand reduction strategies are implemented[27]. Comparison of feasible supply and expected demand gives a sense of the magnitude of the supply-demand gap, which highlights the level of resource efficiency required to close the gap.

## Data availability

The input data and model results of this study have been deposited on GitHub (https://github.com/takumawatari/feasible-material-supply). Permanent references to the data are also accessible through the Zenodo repository[70]. Source data are provided with this paper.

## Code availability

The full model code is available on GitHub (https://github.com/takumawatari/feasible-material-supply). Permanent references to the data are also accessible through the Zenodo repository[70].

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

## Acknowledgements

This research was supported in part by JSPS KAKENHI (21K12344 and 22K18433), the Environment Research and Technology Development Fund (JPMEERF20223001), JST-Mirai Program (JPMJMI21I5), and Engineering and Physical Sciences Research Council (EPSRC) in the United Kingdom through UK FIRES (grant reference EP/S019111/1). We also thank Susannah Dobson for providing helpful comments on the early draft and proofreading.

## Author contributions

T.W.: designed the study, performed the analyses. T.W., A.S., L.G., J.C., and J.A.: prepared the manuscript.

## Competing interests

The authors declare no competing interests.
