## [Peer review file · Nature Communications]

REVIEWER COMMENTS

Reviewer #1 (Remarks to the Author):

The paper departs from the approach to modelling low-carbon scenarios for the steel and cement sector by approaching from the production side. Since the CO₂ budget is limited, it ends up advocating demand reduction but does not clarify strategies to reduce the demand.

The paper also needs to consider what steel and cement supply constraints can mean for both developed and developing countries. Therefore, reliance on zero-emission infrastructure might be a weakness of the conventional approach. This paper proposes an approach which might be politically infeasible.

A number of detailed comments have been provided within the manuscript itself

Reviewer #2 (Remarks to the Author):

The authors provide a valuable contribution to the discussion of the global feasibility of supplying bulk materials within the 1.5C climate budget, with striking simplicity and clarity of their model and exemplary transparency & reproducibility of the underlying code. The authors successfully balance the drawbacks of their simplified model while still drawing meaningful conclusions.

I have the following minor remarks to the paper:

General: I think the authors argument of proposing a “forecasting supply and backcasting demand” instead of the other way around is a key finding of this paper and could be added to the abstract.

General: throughout the paper, the authors continuously mention how the feasible supply within the carbon budget is likely to not meet the future global demand and that risk of deployment failures needs to be anticipated. I somewhat miss the other perspective: that the insufficient infrastructure roll-out might jeopardize our ability to staying within the 1.5C target if we never the less aim to meet the global future supply (and then have to use high-carbon technology routes). This is only implicitly mentioned in the paper.

L. 51.: “however, both of these solutions...”, I struggled to correctly connect “both” with CCUS and hydrogen technologies, as CCUS is mentioned 2 sentences before. Possibly you could consider directly referring to the two solutions by name instead.

L. 60. Could you specify what you mean with “well-established”? Possibly referring to their TLRs as a more systemized way of describing their technological maturity.

L.93-95: could you compare the projected CO₂-capture capacity of steel and cement directly with the actual CO₂-capture capacity of steel and cement, instead of the entire industrial sector? It is not clear to me how much of the 5 Mt is occurring in steel and cement and how much in other industries and what these other industries entail. I have seen that the original source (23) does not provide a further division but maybe secondary literature can clarify that (e.g. <https://status22.globalccsinstitute.com/>): a quick scan of the listed operational CCUS facilities shows that only 0.8Mt/year is occurring in iron and steel, while there are no operational CCUS facilities for cement listed.

L. 110-111 and 375-378: it is not clear to me how you derived the range of emission intensity from the IEA data (fig 1 f) based on your description. Is it based on an assumption of reaching net-zero electricity by 2040 and 2050 and then linearly interpolating from today? If so, I think this is

noteworthy to mention in the text and/or figure description.

L.112-116: Could you provide stronger reasoning why you think it is sufficient to only model low-carbon electricity as a relevant indicator for green hydrogen-related infrastructure? As you point out, the production, storage and transport infrastructure might pose important bottlenecks, possibly more critical than low-carbon electricity.

L. 373-375. Possibly consider adding how you derived the upper ranges, next to the lower ranges, in the figure description for the sake of completeness.

Fig 2: could you elaborate on why the production is first decreasing and then increasing again in your model? This temporal production profile is not sufficiently discussed in the paper in my opinion.

Fig 3: I am not an expert in modelling stocks and flows, but it does strike me as interesting/noteworthy to see such an increase in the Scrap-EAF route, which is mainly attributed (in my possibly limited understanding) to the availability of post-consumer scrap in your model. Similar to the comment right below, I think some more transparency about the underlying parameters in equation 8, their values and sources would be helpful to better understand if the values for the calculated post-consumer scrap and thus, the calculated CapEAF are reasonable.

Methods: While the methods are described very clearly and all input variable are shared openly on Github, I think the method section could still benefit from a table that gives an overview of the variables listed in line 305-313, their values/ranges and the sources of these values, possibly in the appendix, similar to Table S1 and S2. This will help other researchers to better understand the formulas and will also allow them to reuse the data better.

Methods: l.294-300: I think readers without strong understanding of optimization models might struggle to understand how the production shares of the steel production routes are derived. For cement, there is only one. Possible, some additional explanation could be advantageous here.

Table S2: the authors rightfully highlight that the traditional clinker substitutes stemming from fossil-based processes will be less available in the future. As the Clinker-to-cement target is simultaneously expected to drop from 72 to 57 percent by 2050, I would advise the authors to emphasize the uncertainty of the availability of the suggested other SCMs to cover not only the reduced availability of traditional SCMs but also the additional substitution needs.

Table S2: Similarly, the cement industry is currently using a lot of waste materials as thermal fuels, which will become less available in a more circular economy. Similar to the SCMs, I would invite the authors to reflect upon the implication of this reduced availability for the future fuel mix.

Online SI:

- Adding a requirement txt for an environment in which the code runs smoothly would be great!
- Notebook global-cement-model: Typo in header 1 after cell 1

Reviewer #3 (Remarks to the Author):

This work estimates feasible future supply of cement and steel globally under the constraints of a carbon budget for 1.5°C, the amount of carbon captured as well as the availability of renewable electricity. It thereby applies a different approach than it is usually done: instead of modelling how the future supply of materials need to develop to fulfill a certain demand within a carbon budget, it uses potential future infrastructure as a constraint to determine what the maximum supply could be for

steel and cement.

The conclusion is that there might be an undersupply for cement and steel if the infrastructure for green electricity supply and carbon capture and storage facilities are not deployed at very high rates. Thus, the global demand for steel and cement is likely to not be met under the 1.5°C carbon budget.

I recommend the publication of this work as it provides additional insights into the very relevant topic of the future of emission-intensive industries at the example of cement and steel. The applied approach is valuable and offers a reality check of the usual scenario modelling approaches.

The manuscript is well written and results are presented in plots of high quality. The methods seem adequate. The discussion complements the results and puts them into context.

Comments and suggestions for the authors are given below per section. These are primarily minor revisions concerning structure and clarity of the text as well as readability and presentation of the results and methods.

The explanation of the methods and data sources could still be improved to increase clarity and reproducibility. I would recommend reconsidering certain assumptions of the model and adding an additional analysis for the 2°C carbon budget. Suggestions are listed below for the respective sections and the supplementary information. Some of them are categorized as major comments.

It is great to see that the authors aim to publish the entire modelling code as well as data input in a github repository.

Title:

1. The manuscript assesses steel and cement scenarios. The title refers to "bulk materials". I would suggest replacing "bulk materials" with steel and cement in the title, since bulk materials could comprise more materials than only steel and cement.

Abstract:

2. The abstract could add the information of the 1.5°C carbon budget.

3. L. 20 – 22. It is referred to "the expected demand", while there is not only 1 expected demand, but the demand highly depends on the scenario. This aspect could be included in the abstract.

Introduction:

4. L. 48: "The same applies to": it is not clear what is meant here. The previous sentence mentions 2 numbers (40% for steel, 60% for cement). Does academic literature state exactly the same numbers, which are actually averages? Which numbers apply to "the entire industrial sector"?

5. L. 51: Hydrogen is usually not applicable to the cement sector, but only to future steel production. This could be better differentiated in the introduction.

6. L. 51 – 52: "share one thing in common": this expression could be improved.

7. L. 53 – 56: As far as I know, current carbon capture and storage technologies mostly use natural gas as main energy carrier. For the cement sector, the main decarbonization option is CCS, as it is very difficult to electrify or to use hydrogen. For the steel sector, electrification (or increasing secondary production which uses electrified processes) and the use of green hydrogen produced from renewable electricity are the key measures to realize deep decarbonization, even though the IEA scenarios shown in figure S1 seem to have a different opinion for steel. Thus, the two sectors might require different infrastructure needs: CCS mainly for cement versus electricity and green hydrogen primarily for steel. If the authors agree with this view, this distinction could be made a bit clearer in

the introduction.

8. L. 59: what does "the empirical data" refer to?

9. L. 60: "well established": what exactly is meant by this? Would "mature technology" describe the idea better?

10. L. 61: "these technologies must undergo a period...": I would suggest being a bit more precise in this sentence, e.g. "The implementation/deployment of these technologies require"

11. L. 77-78: If I understand correctly, the infrastructure for green hydrogen, e.g. capacity of electrolyzers is not considered in the model, but only capacity of renewable electricity generation. It would be very helpful to the reader to be specific what exactly is modelled and how this is used for the interpretation or role for the respective sectors. Thus, I would recommend adding a short definition of "zero emissions infrastructure" in the introduction (or elsewhere) which explains the meaning of the term for the model used in this work.

Section 2: uncertain infrastructure deployment:

12. Paragraph l. 105 – l. 116: This paragraph could be introduced a bit better. The overall context of figure 1 c-f is not very clear. It could benefit from a general statement about the analysis of electricity generation capacity. The two paragraphs before are well phrased, focusing on CCUS.

13. L. 107: "therefore, while ... ": please improve the language here.

14. L. 109: it would be helpful to add here the details how the linear extrapolation was done.

15. L. 110: is the electricity supply for all sectors or only for cement and steel? Please also specify in Fig. 1d), e)

16. L. 113: if hydrogen is only used for steel and not for cement, it would be helpful to mention this here again.

Major comments:

17. The current approach focuses on the uncertainty in the constraints of captured carbon and the electricity supply. However, also for the carbon budget uncertainty plays a crucial role, e.g. in the maximum temperature increase and the associated probability. The current approach does not include any uncertainty for the carbon budget, but chooses the 1.5°C target and a 50% probability. Generally, it is not considered very likely anymore that the 1.5°C target will be met. It could be very interesting to add an additional analysis which acknowledges the uncertainty for the carbon budget as well, e.g. for the 2°C target. The respective results could be represented as in fig. 2, e.g. in the supplementary information.

18. Paragraph L. 118 – 128: This paragraph summarizes the methods at a very high level. Currently, it does not provide sufficient detail to understand what the optimization model does. I would appreciate one or two additional sentences addressing for example the following questions:

i) What does the model optimize, what is maximized or minimized?

ii) What exactly are the constraints of the model?

iii) Which of the data presented in figure 1 is used in the model and how?

iv) Which carbon budget is used and how is it defined?

v) What exactly is meant with "technological progress" in line 121 and how is this part of the model?

vi) Is the analysis done at a global level without any regionalization? If so, please add this information.

19. L. 123-125: this sentence is very important to understand the main goal of the study. It could be positioned more prominently in the text, e.g. at the beginning of the paragraph instead of in the middle.

20. L. 127: uniform distribution: in Fig. 1d) a trend is clearly visible which is that electricity supply increases with the publication year of the IEA reports, such that the more recent the report, the higher the estimated needed electricity supply. A similar but weaker trend is visible in figure 1b) for CCUS capacity. Could these trends be reflected in the model, e.g. through another distribution than a uniform distribution, such as a beta distribution?

21. L. 127: The term variables is mentioned, but it is not specified what the variables are, e.g. by referring to figure 1.

Figure 1:

22. Fig. 1: if all the data is global, please add this information somewhere.

23. Fig. 1a) – c): Why do the subfigures show the cumulative values for cement and steel instead of presenting them separately, e.g. as separate bars next to each other for each IEA report? Wouldn't this align with the method as well, where the CCS capacities are assessed for each sector individually?

24. Fig. 1c): Would it be possible to also show here the areas separately for cement and steel, instead of cumulatively? This would allow to see the min. and max. value for cement as well.

25. Fig. 1c): Why is the max. value in 2050 considerably lower than in Fig. 1b)?

26. Fig. 1c): Why does steel have another colour than in fig. 1a and b)?

27. Fig. 1c): What do the 2 bars on the right mean?

28. Fig. 1f): How were the values for this figure derived?

29. Fig. 1 caption states an important assumption (l. 375 – 378), which would be great to move or add it to the main text.

30. L. 377: "can increase proportionally to the total electricity supply". What exactly does this mean? That it can increase at the same rate as the total (i.e. global) electricity supply? Does this assumption make sense for the cement sector which is very difficult to electrify while other sectors are more important to electrify? Shouldn't the cement sector rather receive less electricity?

Figure 2:

31. It could be made clearer what the results of the optimization model are and what the estimations according the IEA scenarios are.

Regarding the interpretation and explanation of the figure, adding information regarding the following questions could provide more insights:

i) Why is the uncertainty for cement supply so much larger than for steel?

ii) Why is the uncertainty for cement demand in 2050 so much smaller than for steel?

iii) Is it possible to say something about what the most limiting factors are for the supply of steel versus cement: electricity supply, CCS capacity or the carbon budget?

iv) It is surprising that the curves for both cement and steel first go down and then around 2040 or so exhibit an increasing trend again. How can this be explained?

Section "Certain growth of recycling in an uncertain future"

32. This section is only about steel, which is not obvious from the heading. The heading could be rephrased to communicate this.

33. For steel, CCS is usually applied to fossil-based production technologies, such as BF-BOF or Fossil DRI-EAF. However, their production share is very low in 2050, while the CO₂ captured seems to potentially further increase according to Fig. 1c). Thus, the question arises whether the results for carbon captured from steel production in 2050 practically make sense, or would they imply that CCS is also applied to H₂-DRI or scrap-EAF?

Figure 3:

34. Fig. 3b): the trajectory of fossil DRI-EAF is not explained. It seems that it relates to figure S3 in the supplementary information. Why does it not show higher uncertainties?

Section "Sufficient feasible supply to meet basic human needs"

35. L. 190: what about uncertainty in this data from reference 27? What are the assumptions regarding the development of the world population? Do these not vary depending on scenarios?

36. L. 400 – 404: this seems to be an important information. It could be valuable to move it out of the figure caption into the main text.

37. L. 402: What is the definition of the "required level"?

38. L. 400 – 404: What does this fact imply? It seems to influence the interpretation of the results,

especially for cement where recycling is not common practice. Does this mean that the difference between the feasible supply and the min. requirements for basic human needs could be interpreted smaller than shown in fig. 4?

Discussion:

39. L. 222-224: what about the challenge of exceeding the carbon budget? If industry does not receive the required infrastructure for low-carbon production technologies, they might continue using the current production technologies to satisfy the demand, which will lead to higher emissions than allowed by the carbon budget.

40. L. 237 – 238: It seems to me that this sentence belongs to the next paragraph which describes the demand reduction strategies.

41. L. 240 – 255: The strategies described in this paragraph are great suggestions. Currently, it is phrased in a neutral way such that it is not clear how these strategies could be best realized or incentivized. To which extent is it the responsibility of governments to define respective policies or regulations? If it is their responsibilities, this could be added as a challenge in line 223.

42. L. 288: How can bulk materials be part of the solution to the ongoing climate crisis? This conclusion seems to be not supported by the results presented in the paper.

43. The results of this manuscript are worrying as they suggest that there might be a shortage of steel and cement supply in the future under the 1.5° carbon budget. Based on the results and possible measures presented in the discussion, is it possible to say something about which measures might be the most effective to minimize the supply-demand-gap?

Methods:

Model overview:

44. L 297: please specify the “zero-emissions infrastructure”

45. Equation 2: $CI_i(t)$ seems to be not defined. Please add a definition in form of an equation. Does this include also the emission intensity of electricity from fig. 2f)?

46. It would be very helpful to the reader to add references for the data presented in figure 2 to the respective variables in this section.

47. Equation 4: Cap_{FDRI} is not explained. Does it relate to figure S3 in the supplement?

48. Equations 6-8: It would be valuable to add a flowchart of the system explaining the definition of the different variables used in these equations. This could go in the supplementary information.

49. L. 319: Please correct the grammar of this sentence.

50. L. 326-327: Please refer to the variables here as defined in the equations. “Carbon capture capacity” does not fit to the variable names mentioned for CCS.

51. The methods currently barely name any data sources or quantitative assumptions. Some are mentioned in the main part of the manuscript, some in the supplementary information, some on the github repository. Please explicitly state data sources and assumptions in the methods section or refer to the respective alternative position where they may be specified.

Carbon budgets:

52. L. 336: What are the annual emission mitigation rates. Could they be specified more in detail or a reference added?

Supplementary information:

Table S1:

53. What is the role of the “advanced BF-BOF”, it is not mentioned in the main part of the manuscript.

54. I suppose that the electricity consumption includes the electricity for hydrogen electrolysis. If so,

please add this information.

55. The electricity consumption of H₂-DRI-EAF can largely vary depending on the assumptions of the technology (Bhaskar et al. 2020, Figure 12). What are your technology assumptions, e.g. efficiency of the electrolyzer?

Table S2:

56. How are the target values used in the model? Do the 4 parameters change over time? If so, how? This is not mentioned in the manuscript. Why are such individual improvements assumed for cement, but not for steel, where e.g. efficiency improvements could also apply?

Table S3:

57. This table is neither referred to in the manuscript, nor is it explained how its data is used in the model.

58. Please add an explanation for how the model incorporates energy penalty for CCS.

59. Which energy carriers and resulting emissions are assumed for the energy requirement for CCS?

60. Why is the energy penalty for steel CCS so low. Which assumptions are taken for this technology? Depending on the choice of CCS technology, the current assumption seems too low (Perpinan et al. 2023, figure 16).

Figure S3:

61. This figure is not mentioned or explained in the manuscript. What is the reasoning behind the choice of the values illustrated in the figure? Is the trajectory not subject to uncertainty?

References

Bhaskar, A., Assadi, M., & Nikpey Somehsaraei, H. (2020). Decarbonization of the iron and steel industry with direct reduction of iron ore with green hydrogen. *Energies*, 13(3), 758.

Perpiñán, J., Peña, B., Bailera, M., Eveloy, V., Kannan, P., Raj, A., ... & Romeo, L. M. (2023). Integration of carbon capture technologies in blast furnace based steel making: A comprehensive and systematic review. *Fuel*, 336, 127074.

Response to reviewers

Reviewer #1 (Remarks to the Author):

Comment

The paper departs from the approach to modelling low-carbon scenarios for the steel and cement sector by approaching from the production side. Since the CO2 budget is limited, it ends up advocating demand reduction but does not clarify strategies to reduce the demand.

>Response

Thank you for dedicating your valuable time to reviewing our paper. We sincerely appreciate your feedback and comments. The primary focus of our study is to provide a clear signal to stakeholders about the urgency of implementing demand-side actions. This is why we have adopted a 'forecasting supply and backcasting demand' modeling approach instead of the traditional 'forecasting demand and backcasting supply' approach. By examining the feasible supply and exploring how to use it on the demand side, we aim to emphasize the critical need for demand reduction strategies.

We have dedicated the discussion section to present various demand reduction options and their potential. These options are based on extensive research conducted in our group over the past decade, which allows us to provide informed insights into the possibilities associated with demand reduction strategies. We hope this clarification addresses your concerns and demonstrates the rationale behind our chosen approach.

Comment

The paper also needs to consider what steel and cement supply constraints can mean for both developed and developing countries. Therefore, reliance on zero-emission infrastructure might be a weakness of the conventional approach. This paper proposes an approach which might be politically infeasible.

>Response

This paper has a section entitled "Inequality as a major challenge to meeting basic human needs" which specifically discusses this point. We struggle to fully understand the intent of the latter part of this comment.

Comment

L. 98: The current plans are only till 2030 whereas the scenarios projections are for 2050.

L. 105: IEA numbers are scenario projections and not forecasts that should be compared with current achievement. It may help to put a few lines to explain why CCUS is not happening at scale.

>Response

Thank you for these comments. We think it is important to compare the IEA scenarios with current achievements and learn from them in terms of the challenges we face. Therefore, we have not changed our story in this domain.

Comment

L. 108: It is not clear from 1c what the lower bound is. Therefore revise the figure.

>Response

Thank you for emphasizing this important point. We have revised the Fig. 1c accordingly.

Comment

L. 121: Do you mean the technology shares of the steel and cement technologies were decided exogenously. Since you are using an optimisation why is it not based on the overall system costs. Please also provide the costs of these different steel and cement making technologies and how these costs are expected to go down over time. In steel making you have also included Scrap-EAF and have some constraints been put for this to keep in mind the scrap availability. Also check the approach for modelling steel scrap in Dhar, S., M. Pathak, and P.R. Shukla. 2020. "Transformation of India's Steel and Cement Industry in a Sustainable 1.5 °C World." *Energy Policy* 137. <https://doi.org/10.1016/j.enpol.2019.111104>.

L. 131: In an optimisation model demand for steel and cement is exogeneously and the how this demand will be met is decided by the model. Therefore the point being made that there is a gap between production and demand is not clear at all.

L. 168: The total steel demand is IEA Net Zero Scenario is much higher than what is the result of the optimisation result presented in Fig 3. Therefore is steel demand varied for this paper in the different runs.

>Response

Thank you for the opportunity to clarify these points. What is exogenously determined in our model is the energy performance of each technology, while the share of production technology (e.g., H2 DRI-EAF) is endogenously determined in the model. We deliberately avoid cost optimization with given demand, because we believe that a 'forecasting demand and backcasting supply' approach has not provided the relevant stakeholders with clear signals for demand-side actions. What we propose in this study is a 'forecasting supply and backcasting demand' approach, where we can highlight the urgent necessity of demand-side actions by first better understanding the feasible supply and then exploring how to use it on the demand side. The proposed approach provides clearer evidence and incentives for related stakeholders to take action by demonstrating the limited material availability. This point has been elaborated on in the discussion section. Our model is based on a physical mass balance and the availability of scrap is explicitly considered, as described in the methods section. We have added the following figure to clarify this point:

Supplementary Fig. 2 Conceptual flow chart of iron and steel flows. The symbols at the bottom right of the process show the associated coefficients; the three equations show scrap estimation formulas based on simple mass balance.

Comment

L. 184: The approach for determining minimum needs some explanation. It is not clear from what is written here. Plus the concept is a little problematic since the minimum needs for developed and developing countries will be quite different since in low and middle income countries a lot of infrastructures have to be still created.

>Response

The estimates of minimum requirements are based on a previous study, and this fact is clearly stated in our manuscript. The estimates are also based on differences in infrastructure levels between developed and developing countries, as shown in Fisch-Romito, V. (2021).

Comment

L. 205: A substantial part of material demand also arises from stock replacement and therefore developed countries may continue to use a lot of materials. You can refer to this aspect as well from literature: Allwood, J.M., Ashby, M.F., Gutowski, T.G., Worrell, E., 2011. Material efficiency: a white paper. *Resour. Conserv. Recycl.* 55, 362–381. <https://doi.org/10.1016/J.RESCONREC.2010.11.002>. Allwood, J.M., Cullen, J.M., Milford, R.L., 2010. Options for achieving a 50% cut in industrial carbon emissions by 2050. *Environ. Sci. Technol.* 44, 1888–1894. <https://doi.org/10.1021/es902909k> Dhar, S., M. Pathak, and P.R. Shukla. 2020. “Transformation of India’s Steel and Cement Industry in a Sustainable 1.5 °C World.” *Energy Policy* 137. <https://doi.org/10.1016/j.enpol.2019.111104>.

>Response

We thank the reviewer for highlighting this important point. We have revised the text as follows to clarify this point:

“However, fundamental challenges are posed by the ever-growing needs of high-income countries and the ‘equitable’ distribution of the feasible supply of steel and cement. As shown in **Fig. 5**, current per capita material stocks in high-income countries far exceed the feasible stock levels

derived from the 1.5 °C budget, while low-income countries have material stocks well below the feasible levels. However, it is possible to meet basic human needs across the world population with the feasible stock levels. The challenge lies in the substantial material demand generated by high-income countries for stock replacement and expansion, with houses and cars becoming larger and heavier year after year ²⁷. These trends suggest that inequalities in the use of steel and cement, if not addressed head-on, could impede the provision of the basic needs of the global population within the feasible supply. Since more than 90% of material requirements to meet basic human needs will come from lower-middle and low-income countries, predominantly in South Asia and sub-Saharan Africa, the primary challenge will be to balance the ever-growing demands of high-income countries ²⁸, with the need to distribute the feasible supply equitably.”

Comment

L. 222: Recycling of steel e.g., as scrap still needs energy albeit lower and can potentially lead to CO2 emissions. Maybe reuse, dematerialization is what could be proposed.

>Response

We focus specifically on the strategy of the materials industry in this text. Reuse and dematerialization are therefore not consistent with the context.

Comment

L. 259: The paper is advocating demand reduction but not clarifying the strategies that can help in reducing the demand.

>Response

Our manuscript discusses the demand reduction strategies based on our extensive research in this area over the decades:

“Our analysis provides clear signals for the related stakeholders to promote a suite of demand-side resource efficiency strategies ³². For example, there is an important opportunity to almost halve the use of materials in the building frame through optimized design and reduced overdesign ³³. Strategic urban planning that encourages the shift from private cars to public transport can reduce the material use per distance traveled ³⁴. Upfront design for material utilization and flexible blanking equipment can significantly reduce material losses in manufacturing ³⁵. Material loss during construction can also be reduced by improving the architectural and engineering specifications and addressing a lack of client interest ³⁶. Products can last longer through changing consumer behaviors ³⁷. There is also evidence that a large portion of end-of-life material components can be reused without melting down if we have the will to do so ³⁸. Existing evidence suggests that the construction and manufacturing industries can provide the same level of services with around 35-70% and 55-65% less material use, respectively (**Supplementary Tables 5 and 6**).”

Reviewer #2 (Remarks to the Author):

Comment

The authors provide a valuable contribution to the discussion of the global feasibility of supplying bulk materials within the 1.5C climate budget, with striking simplicity and clarity of their model and exemplary transparency & reproducibility of the underlying code. The authors successfully balance the drawbacks of their simplified model while still drawing meaningful conclusions. I have the following minor remarks to the paper:

>Response

Thank you very much for your valuable time spent reading our paper. We greatly appreciate the reviewer's constructive and insightful feedback. We have endeavored to revise the paper accordingly and we believe that this has resulted in significant improvements. Please see our point-by-point responses and the revisions that we have made to the manuscript.

Comment

General: I think the authors argument of proposing a “forecasting supply and backcasting demand” instead of the other way around is a key finding of this paper and could be added to the abstract.

>Response

Thank you for highlighting this point. We agree with the reviewer and have added the following text to the abstract:

“Our feasible supply scenarios provide compelling evidence of the urgency of demand-side actions and establish benchmarks for the required level of resource efficiency.”

Comment

General: throughout the paper, the authors continuously mention how the feasible supply within the carbon budget is likely to not meet the future global demand and that risk of deployment failures needs to be anticipated. I somewhat miss the other perspective: that the insufficient infrastructure roll-out might jeopardize our ability to staying within the 1.5C target if we never the less aim to meet the global future supply (and then have to use high-carbon technology routes). This is only implicitly mentioned in the paper.

>Response

We greatly appreciate this insightful comment. Indeed, our manuscript did not carefully emphasize this point. We have added the following discussion to underline this point:

*“If these challenges are not addressed head-on, the gap between feasible supply and expected demand risks being filled by emission-intensive production processes. This could result in cumulative emissions of ~160 Gt-CO₂ by 2050, representing ~40% of the remaining 1.5°C budget or ~20% of the remaining well-below 2°C budget (**Supplementary Fig. 6**). Avoiding this future depends on how the current generation prepares for an uncertain future; we must not leave it to the enormous efforts of future generations.”*

Comment

L. 51.: “however, both of these solutions...”, I struggled to correctly connect “both” with CCUS and hydrogen technologies, as CCUS is mentioned 2 sentences before. Possibly you could consider directly referring to the two solutions by name instead.

>Response

We apologize for our unclear sentence. We have revised it as follows:

“However, both CCUS and hydrogen-based solutions share a common challenge: the indispensability of infrastructure.”

Comment

L. 60. Could you specify what you mean with “well-established”? Possibly referring to their TLRs as a more systemized way of describing their technological maturity.

>Response

Thank you for this comment. We have revised the text as follows:

“CCUS and green hydrogen production are mature technologies with a high level of technology readiness ¹¹, but they have not yet been deployed at scale.”

Comment

L.93-95: could you compare the projected CO₂-capture capacity of steel and cement directly with the actual CO₂-capture capacity of steel and cement, instead of the entire industrial sector? It is not clear to me how much of the 5 Mt is occurring in steel and cement and how much in other industries and what these other industries entail. I have seen that the original source (23) does not provide a further division but maybe secondary literature can clarify that (e.g. <https://status22.globalccsinstitute.com/>): a quick scan of the listed operational CCUS facilities shows that only 0.8Mt/year is occurring in iron and steel, while there are no operational CCUS facilities for cement listed.

>Response

We thank the reviewer for this insightful comment. Fortunately, the latest IEA database provides the separate data for the steel and cement sectors. Thus, it is now possible to directly compare the IEA scenario projections with the current situation. The revised text is as follows:

*“First, current CCUS capacity falls short of the levels that past IEA reports assumed would be deployed by 2021 (**Fig. 1a**). For instance, the 2010 IEA report assumed that CO₂ capture for the steel and cement sectors would reach ~195 Mt-CO₂ in 2021 but the current operating capacity for the steel and cement sectors is just under 1 Mt-CO₂ ¹⁷. It appears that CCUS-related infrastructure has not been deployed as originally planned. Second, the 2050 CCUS capacity envisaged in the IEA scenarios requires an expansion at a rate that far exceeds current construction plans (**Fig. 1b**). Despite the historical failure of CCUS deployment, the IEA scenarios consistently assume ~2000 Mt-CO₂ capture in the steel and cement sectors for 2050, which is 2000 times the current capacity for these sectors (~1 Mt-CO₂) and more than 100 times the 2030 construction plan (~19 Mt-CO₂) ¹⁷.”*

Comment

L. 110-111 and 375-378: it is not clear to me how you derived the range of emission intensity from the IEA data (fig 1 f) based on your description. Is it based on an assumption of reaching net-zero electricity by 2040 and 2050 and then linearly interpolating from today? If so, I think this is noteworthy to mention in the text and/or figure description.

>Response

We apologize for our unclear explanation. We have revised the text to clarify this point:

“A similar approach is taken for non-emitting electricity supply. The IEA scenarios tend to assume higher levels of total non-emitting electricity supply in more recent reports (Fig. 1d). This may reflect two factors: the success of cost reductions in solar and wind, and the failure of emission reductions, leading to more stringent electrification requirements now and in the future¹⁹. However, despite the substantial expansion of non-emitting electricity supply, it still falls short of the annual increase needed to meet the most ambitious scenario²⁰. This study assumes that the scenario ranges reflect the uncertainty associated with the future deployment of non-emitting electricity. Therefore, the upper and lower bounds for both electricity supply and its emission intensity are established based on the IEA scenario ranges (Fig. 1e and 1f).”

Comment

L.112-116: Could you provide stronger reasoning why you think it is sufficient to only model low-carbon electricity as a relevant indicator for green hydrogen-related infrastructure? As you point out, the production, storage and transport infrastructure might pose important bottlenecks, possibly more critical than low-carbon electricity.

>Response

Thank you for this extremely insightful comment. We completely agree with the reviewer and have revised the text as follows:

“Our analysis specifically focuses on electricity supply as a relevant indicator of green hydrogen uptake, although various infrastructure components (i.e., electrolyzers, storage tanks, and transport facilities) also need to be built. This assumption is grounded in the understanding that affordable and reliable electricity is a fundamental requirement for the competitiveness of green hydrogen-based DRI⁹. Consequently, we posit that the expansion of green hydrogen production can align with the rate of electricity expansion.”

Comment

L. 373-375. Possibly consider adding how you derived the upper ranges, next to the lower ranges, in the figure description for the sake of completeness.

>Response

Thank you for this comment. We have modified the figure caption accordingly.

Comment

Fig 2: could you elaborate on why the production is first decreasing and then increasing again in your model? This temporal production profile is not sufficiently discussed in the paper in my opinion.

>Response

We thank the reviewer for highlighting this important point. We have added the following text to clarify this point:

“A comparison of steel and cement highlights three caveats. First, the uncertainty in cement supply is greater than the uncertainty in steel supply. This is due to the relatively limited options for decarbonizing the cement production process, which depends exclusively on the deployment of CCUS. Second, the range of expected demand is greater for steel than for cement. This discrepancy likely stems from the limited exploration of demand reduction strategies for cement. In fact, the LED scenario²⁴ only considers one strategy for cement due to the lack of scientific

evidence, while a broader range of strategies are examined for steel. Third, the feasible supply of both steel and cement exhibits an initial downward trend, followed by an upward trend. This pattern is attributed to the combined effects of the shape of the carbon budget, which requires immediate emission reductions, and the linear deployment of zero-emission infrastructure, which is assumed to increase steadily over time.”

Comment

Fig 3: I am not an expert in modelling stocks and flows, but it does strike me as interesting/noteworthy to see such an increase in the Scrap-EAF route, which is mainly attributed (in my possibly limited understanding) to the availability of post-consumer scrap in your model. Similar to the comment right below, I think some more transparency about the underlying parameters in equation 8, their values and sources would be helpful to better understand if the values for the calculated post-consumer scrap and thus, the calculated CapEAF are reasonable.

>Response

Thank you for this insightful feedback. Indeed, our previous manuscript was not clear enough on this point. We have added the following two figures accordingly:

Supplementary Fig. 2 Conceptual flow chart of iron and steel flows. The symbols at the bottom right of the process show the associated coefficients; the three equations show scrap estimation formulas based on simple mass balance.

Supplementary Fig. 5 Recovery rate of post-consumer scrap. The historical data are calculated based on Ref ¹¹ and the 2050 target value is based on Ref ¹⁰.

Comment

Methods: While the methods are described very clearly and all input variable are shared openly on Github, I think the method section could still benefit from a table that gives an overview of the variables listed in line 305-313, their values/ranges and the sources of these values, possibly in the appendix, similar to Table S1 and S2. This will help other researchers to better understand the formulas and will also allow them to reuse the data better.

>Response

Thank you for this comment. We have added the table summarizing all input variables and the sources of these values:

Supplementary Table 1 Summary of system variables and parameters.

Symbol	Description	Data
P_i	Material production in route i	Optimization variables
$CI_i^{(f)}$	Emission intensity of material production in route i due to fuel combustion and chemical process.	Tables S1 and S2
EI_i	Electricity intensity of material production in route i	Tables S1 and S2
$CI^{(e)}$	Emission intensity of electricity use	Fig. 1f in the main text
CC	Captured carbon through CCUS	Fig. 1c in the main text
EI_{cc}	Energy penalty of carbon capture technologies	Table S3
Cap_{Carbon}	Carbon budget	Figure S3
$Cap_{Electricity}$	Maximum electricity supply	Fig. 1e in the main text
Cap_{F_DRI}	Maximum production via fossil DRI-EAF route	Figure S4
Cap_{EAF}	Maximum production via scrap-EAF route	Mass-balancing

Pre_{EAF}	Recovered pre-consumer scrap for use in scrap-EAF route	Mass-balancing
$Post$	Recovered post-consumer scrap	Mass-balancing
θ	EAF production yield	96% (Ref ¹⁰)
δ	Forming yield	93% (Ref ¹¹)
λ	Fabrication yield	86% (Ref ¹¹)
ω	Ratio of scrap in total BOF input	20% (Ref ¹⁰)
γ	Recovery rate of post-consumer scrap	Figure S5 (calculated based on Refs ^{10,11})
ρ	Hibernation ratio of end-of-life products	15% (Ref ¹²)
ϕ	Lifetime distribution	Note

Note: We assume a normal distribution with a mean of 38 years for steel and 50 years for cement and a standard deviation of 30% of the mean ^{13–15}.

Comment

Methods: I.294-300: I think readers without strong understanding of optimization models might struggle to understand how the production shares of the steel production routes are derived. For cement, there is only one. Possible, some additional explanation could be advantageous here.

>Response

Thank you for emphasizing this point. We have added the following text:

“The deployment of each production process is selected endogenously according to its profile regarding carbon emissions, electricity use, and resource requirements.”

Comment

Table S2: the authors rightfully highlight that the traditional clinker substitutes stemming from fossil-based processes will be less available in the future. As the Clinker-to-cement target is simultaneously expected to drop from 72 to 57 percent by 2050, I would advise the authors to emphasize the uncertainty of the availability of the suggested other SCMs to cover not only the reduced availability of traditional SCMs but also the additional substitution needs.

Table S2: Similarly, the cement industry is currently using a lot of waste materials as thermal fuels, which will become less available in a more circular economy. Similar to the SCMs, I would invite the authors to reflect upon the implication of this reduced availability for the future fuel mix.

>Response

We thank the reviewer for these insightful comments. We have added the following explanation in the main text to highlight these points appropriately:

“In this domain, four key assumptions deserve attention. First, the hydrogen-based direct reduction system is based on the literature ⁵⁴ with an electrolyzer efficiency of 45 kWh/kg H₂ and a hydrogen mass flow rate of 1.5 (i.e., 50% oversupply of hydrogen for full conversion of iron ore in the shaft ⁸). Since this electrolyzer efficiency is already at a high level ⁶³, no further efficiency improvement is assumed. Second, clinker has traditionally been substituted mainly as fly ash and granulated blast furnace slag (GBFS), which will decline in a decarbonized future ⁶⁴. Therefore, clinker substitution will have to be provided as resources other than fly ash and GBFS. Our assumption here is based on recent studies that show significant potential for substitution with

calcined clay, agricultural by-product ash, forestry by-product ash, and end-of-life binders ^{65,66}. *Third, similar to supplementary cementitious materials, the cement industry currently uses waste materials as thermal fuels, which may become less available in a decarbonized, more circular future* ⁵². *We assume an expanded use of waste from agricultural, chemical, and food production* ⁵⁶. *It is important to note that a more comprehensive assessment of resource availability needs to consider the interconnected system beyond just the steel and cement industries* ⁶⁷. *Fourth, technological improvements are assumed to be achieved linearly over the period 2050. This assumption, although simplistic, is based on the limited evidence and is consistent with industry roadmaps that occasionally assume linear progress* ⁵⁶.”

Comment

Online SI:

- Adding a requirement txt for an environment in which the code runs smoothly would be great!
- Notebook global-cement-model: Typo in header 1 after cell 1

>Response

Thank you for bringing these points to our attention. We have added the requirements.txt file and modified the typo.

Reviewer #3 (Remarks to the Author):

Comment

This work estimates feasible future supply of cement and steel globally under the constraints of a carbon budget for 1.5°C, the amount of carbon captured as well as the availability of renewable electricity. It thereby applies a different approach than it is usually done: instead of modelling how the future supply of materials need to develop to fulfill a certain demand within a carbon budget, it uses potential future infrastructure as a constraint to determine what the maximum supply could be for steel and cement. The conclusion is that there might be an undersupply for cement and steel if the infrastructure for green electricity supply and carbon capture and storage facilities are not deployed at very high rates. Thus, the global demand for steel and cement is likely to not be met under the 1.5°C carbon budget.

I recommend the publication of this work as it provides additional insights into the very relevant topic of the future of emission-intensive industries at the example of cement and steel. The applied approach is valuable and offers a reality check of the usual scenario modelling approaches. The manuscript is well written and results are presented in plots of high quality. The methods seem adequate. The discussion complements the results and puts them into context.

Comments and suggestions for the authors are given below per section. These are primarily minor revisions concerning structure and clarity of the text as well as readability and presentation of the results and methods. The explanation of the methods and data sources could still be improved to increase clarity and reproducibility. I would recommend reconsidering certain assumptions of the model and adding an additional analysis for the 2°C carbon budget. Suggestions are listed below for the respective sections and the supplementary information. Some of them are categorized as major comments. It is great to see that the authors aim to publish the entire modelling code as well as data input in a github repository.

>Response

Thank you very much for your valuable time spent reading our paper. We are extremely grateful for the very detailed and insightful feedback. We have endeavored to revise the paper accordingly and we believe that this has resulted in significant improvements. Please see our point-by-point responses and the revisions that we have made to the manuscript.

Comment

Title:

1. The manuscript assesses steel and cement scenarios. The title refers to “bulk materials”. I would suggest replacing “bulk materials” with steel and cement in the title, since bulk materials could comprise more materials than only steel and cement.

>Response

We agree with the reviewer and have revised the title accordingly.

Comment

Abstract:

2. The abstract could add the information of the 1.5°C carbon budget.

>Response

Thank you for raising this important point. We have revised the abstract to use the phrase "Paris-compliant carbon budgets", borrowing a phrase from the paper: Calverley, D., & Anderson, K. (2022). Phaseout Pathways for Fossil Fuel Production Within Paris-compliant Carbon Budgets. <https://www.iisd.org/publications/report/phaseout-pathways-fossil-fuel-production-within-paris-compliant-carbon-budgets>

Comment

3. L. 20 – 22. It is referred to “the expected demand”, while there is not only 1 expected demand, but the demand highly depends on the scenario. This aspect could be included in the abstract.

>Response

Thank you for this comment. We have added the "baseline" to the text to emphasize this point. We would have liked to explain more in the abstract, but due to strict word limits, we had to limit our revisions to minor ones. We hope this minor revision still makes the point clear.

Comment

Introduction:

4. L. 48: “The same applies to”: it is not clear what is meant here. The previous sentence mentions 2 numbers (40% for steel, 60% for cement). Does academic literature state exactly the same numbers, which are actually averages? Which numbers apply to “the entire industrial sector”?

>Response

Thank you for bringing this to our attention. We have revised the text as follows:

“This trend is also reflected in the academic literature on steel⁵, cement⁶, or the entire industrial sector⁷, where a significant part of the emission reductions are expected to come from CCUS.”

Comment

5. L. 51: Hydrogen is usually not applicable to the cement sector, but only to future steel production. This could be better differentiated in the introduction.

>Response

This is indeed an important point. We agree with the reviewer and have added the following text: *“Recent evidence indicates that hydrogen-based steel can be economically competitive when combined with high-quality iron ore, low steelworker wages, and abundant and cost-effective renewable electricity⁹.”*

Comment

6. L. 51 – 52: “share one thing in common”: this expression could be improved.

>Response

Thank you for highlighting this point. We have revised the text as follows:

“However, both CCUS and hydrogen-based solutions share a common challenge: the indispensability of infrastructure.”

Comment

7. L. 53 – 56: As far as I know, current carbon capture and storage technologies mostly use natural gas as main energy carrier. For the cement sector, the main decarbonization option is CCS, as it is very difficult to electrify or to use hydrogen. For the steel sector, electrification (or increasing

secondary production which uses electrified processes) and the use of green hydrogen produced from renewable electricity are the key measures to realize deep decarbonization, even though the IEA scenarios shown in figure S1 seem to have a different opinion for steel. Thus, the two sectors might require different infrastructure needs: CCS mainly for cement versus electricity and green hydrogen primarily for steel. If the authors agree with this view, this distinction could be made a bit clearer in the introduction.

>Response

Thank you for this useful suggestion. We have added the following text to clarify this point:

“In particular, steel decarbonization benefits from all of these technologies, while cement decarbonization relies heavily on CCUS.”

Comment

8. L. 59: what does “the empirical data” refer to?

>Response

Thank you for this question. We have replaced the word with “historical data”.

Comment

9. L. 60: “well established”: what exactly is meant by this? Would “mature technology” describe the idea better?

>Response

We apologize for the unclear text in the earlier manuscript. We have revised the text as follows:

“CCUS and green hydrogen production are mature technologies with a high level of technology readiness¹¹, but they have not yet been deployed at scale.”

Comment

10. L. 61: “these technologies must undergo a period...”: I would suggest being a bit more precise in this sentence, e.g. “The implementation/deployment of these technologies require”

>Response

Thank you for this useful comment. We have revised the text accordingly.

Comment

11. L. 77-78: If I understand correctly, the infrastructure for green hydrogen, e.g. capacity of electrolyzers is not considered in the model, but only capacity of renewable electricity generation. It would be very helpful to the reader to be specific what exactly is modelled and how this is used for the interpretation or role for the respective sectors. Thus, I would recommend adding a short definition of “zero emissions infrastructure” in the introduction (or elsewhere) which explains the meaning of the term for the model used in this work.

>Response

Thank you for highlighting this important point. We have added the following text to clarify this point:

“Specifically, we consider two types of zero-emissions infrastructure: CCUS and non-emitting electricity.”

Comment

Section 2: uncertain infrastructure deployment:

12. Paragraph I. 105 – I. 116: This paragraph could be introduced a bit better. The overall context of figure 1 c-f is not very clear. It could benefit from a general statement about the analysis of electricity generation capacity. The two paragraphs before are well phrased, focusing on CCUS.

13. L. 107: “therefore, while ... ”: please improve the language here.

14. L. 109: it would be helpful to add here the details how the linear extrapolation was done.

>Response

We thank the reviewer for emphasizing this important point. We agree that the explanation in the earlier manuscript was not clear. We have revised the text as follows:

“This is not to say that the IEA scenarios are physically or economically unrealistic, but that their realization is highly uncertain given the scale of the challenge and our historical failures. Therefore, this study makes the following assumptions about CCUS deployment (Fig. 1c): First, the upper bound of CCUS deployment is based on the most ambitious IEA Net Zero scenario¹⁸. This scenario envisions the steel and cement sectors achieving CCUS capacities of 670 Mt-CO₂ and 1355 Mt-CO₂, respectively, by 2050. Second, the lower bound is derived from a more conservative linear extrapolation based on the current operating capacity and the 2030 construction plan. This conservative estimate yields CCUS capacities of 15 Mt-CO₂ and 50 Mt-CO₂ for the steel and cement sectors, respectively, by 2050.

A similar approach is taken for non-emitting electricity supply. The IEA scenarios tend to assume higher levels of total non-emitting electricity supply in more recent reports (Fig. 1d). This may reflect two factors: the success of cost reductions in solar and wind, and the failure of emission reductions, leading to more stringent electrification requirements now and in the future¹⁹. However, despite the substantial expansion of non-emitting electricity supply, it still falls short of the annual increase needed to meet the most ambitious scenario²⁰. This study assumes that the scenario ranges reflect the uncertainty associated with the future deployment of non-emitting electricity. Therefore, the upper and lower bounds for both electricity supply and its emission intensity are established based on the IEA scenario ranges (Fig. 1e and 1f).”

Comment

15. L. 110: is the electricity supply for all sectors or only for cement and steel? Please also specify in Fig. 1d), e)

>Response

Thank you for your question. We have specified in the main text and figure caption that it is the total electricity supply.

Comment

16. L. 113: if hydrogen is only used for steel and not for cement, it would be helpful to mention this here again.

>Response

Thank you for this useful suggestion. We have emphasized this point in the text accordingly.

“This assumption is grounded in the understanding that affordable and reliable electricity is a fundamental requirement for the competitiveness of green hydrogen-based DRI⁹. Consequently, we posit that the expansion of green hydrogen production can align with the rate of electricity expansion.”

Comment

Major comments:

17. The current approach focuses on the uncertainty in the constraints of captured carbon and the electricity supply. However, also for the carbon budget uncertainty plays a crucial role, e.g. in the maximum temperature increase and the associated probability. The current approach does not include any uncertainty for the carbon budget, but chooses the 1.5°C target and a 50% probability. Generally, it is not considered very likely anymore that the 1.5°C target will be met. It could be very interesting to add an additional analysis which acknowledges the uncertainty for the carbon budget as well, e.g. for the 2°C target. The respective results could be represented as in fig. 2, e.g. in the supplementary information.

>Response

We thank the reviewer for this thought-provoking comment. Indeed, the uncertainty in carbon budgets is critically important. Accordingly, in addition to a 1.5°C carbon budget, we have added the analysis for a 1.7°C carbon budget with a 50% probability (equivalent to an 83% probability of 2.0°C), which corresponds to "well-below 2°C" in the Paris Agreement. We hope this additional analysis validates our main messages.

Comment

18. Paragraph L. 118 – 128: This paragraph summarizes the methods at a very high level. Currently, it does not provide sufficient detail to understand what the optimization model does. I would appreciate one or two additional sentences addressing for example the following questions:

- i) What does the model optimize, what is maximized or minimized?
- ii) What exactly are the constraints of the model?
- iii) Which of the data presented in figure 1 is used in the model and how?
- iv) Which carbon budget is used and how is it defined?
- v) What exactly is meant with “technological progress” in line 121 and how is this part of the model?
- vi) Is the analysis done at a global level without any regionalization? If so, please add this information.

>Response

We thank the reviewer for emphasizing several important points about the model. We have revised the text to clarify these points:

*“The potential ranges of zero-emissions infrastructure deployment, as depicted in **Fig. 1c, 1e, and 1f**, are fed into the optimization model. The model attempts to maximize the global supply of steel and cement under the constraints of carbon budgets, infrastructure deployment, and scrap availability through physical mass balancing. To avoid arbitrary judgments about the most likely value of zero-emissions infrastructure deployment or the shape of its distribution, a uniform distribution is used for each variable (i.e., **Fig. 1c, 1e, and 1f**) between its upper and lower bounds²². The carbon budgets employed in the model are based on limiting the global mean temperature*

rise within 1.5 °C with a 50% probability, consistent with the Paris Agreement³. We also consider a carbon budget for a 50% probability of 1.7 °C (equivalent to an 83% probability of 2.0 °C), which corresponds to a well-below 2 °C budget²³. To isolate the impact of infrastructure uncertainty on the supply of steel and cement, we presume that technological progress in these industries will align with established roadmaps and industry-accepted best practices (see Methods section).”

Comment

19. L. 123-125: this sentence is very important to understand the main goal of the study. It could be positioned more prominently in the text, e.g. at the beginning of the paragraph instead of in the middle.

>Response

Thank you for this useful suggestion. We have adjusted the text accordingly.

Comment

20. L. 127: uniform distribution: in Fig. 1d) a trend is clearly visible which is that electricity supply increases with the publication year of the IEA reports, such that the more recent the report, the higher the estimated needed electricity supply. A similar but weaker trend is visible in figure 1b) for CCUS capacity. Could these trends be reflected in the model, e.g. through another distribution than a uniform distribution, such as a beta distribution?

>Response

Thank you for raising this important point. It is indeed an interesting idea to adjust the distributions based on the trend of the IEA scenarios. However, after the internal discussion within the team, we decided to use a uniform distribution for two reasons. First, the trends observed in the IEA scenarios do not necessarily reflect the actual feasibility of the scenarios. Second, the choice of a uniform distribution simplifies the modeling process and increases transparency, which is essential for replicability and to facilitate discussion and further research on the topic. We thank the reviewer for the suggestion. This comment has certainly inspired our next research idea.

Comment

21. L. 127: The term variables is mentioned, but it is not specified what the variables are, e.g. by referring to figure 1.

>Response

Thank you for highlighting this point. We have referred to Fig. 1 in the text.

Comment

Figure 1:

22. Fig. 1: if all the data is global, please add this information somewhere.

>Response

Thank you for emphasizing this point. We have added “global” to the caption.

Comment

23. Fig. 1a) – c): Why do the subfigures show the cumulative values for cement and steel instead of presenting them separately, e.g. as separate bars next to each other for each IEA report?

Wouldn't this align with the method as well, where the CCS capacities are assessed for each sector individually?

24. Fig. 1c): Would it be possible to also show here the areas separately for cement and steel, instead of cumulatively? This would allow to see the min. and max. value for cement as well.

>Response

Thank you for bringing this to our attention. We chose the stacked bar chart because the steel and cement sectors are interrelated in the IEA model, as their cost optimization model considers all sectors together in its estimate, not each sector separately. We were also concerned that presenting separate bars would make the figure too busy to understand. In this regard, we believe it is still appropriate to present the IEA data as a stacked bar chart. On the other hand, as you pointed out, our model treats the steel and cement sectors separately, and we have designed Fig. 1c to show this concept more clearly.

Comment

25. Fig. 1c): Why is the max. value in 2050 considerably lower than in Fig. 1b)?

>Response

Data in Fig. 1c are consistent with those in Fig. 1b.

Comment

26. Fig. 1c): Why does steel have another colour than in fig. 1a and b)?

>Response

We use the same colors but with transparency to make the overlap visible.

Comment

27. Fig. 1c): What do the 2 bars on the right mean?

>Response

The right-hand bars in Fig. 1c show the range of 2050 values.

Comment

28. Fig. 1f): How were the values for this figure derived?

>Response

We apologize for our unclear explanation in the earlier manuscript. We have revised the explanation as described above.

Comment

29. Fig. 1 caption states an important assumption (l. 375 – 378), which would be great to move or add it to the main text.

>Response

This is indeed an important assumption. We have moved this text to the main text. Thank you for pointing this out.

Comment

30. L. 377: "can increase proportionally to the total electricity supply". What exactly does this mean? That it can increase at the same rate as the total (i.e. global) electricity supply? Does this

assumption make sense for the cement sector which is very difficult to electrify while other sectors are more important to electrify? Shouldn't the cement sector rather receive less electricity?

>Response

Thank you for this question. Yes, what we mean here is that the electricity available for both steel and cement production can increase proportionally to the total electricity supply. The text has been revised as indicated above to clarify that this assumption does not have much impact on the cement sector.

Comment

Figure 2:

31. It could be made clearer what the results of the optimization model are and what the estimations according the IEA scenarios are.

>Response

Thank you for bringing this point to our attention. We have revised the figure to clarify our results and the IEA scenarios.

Comment

Regarding the interpretation and explanation of the figure, adding information regarding the following questions could provide more insights:

- i) Why is the uncertainty for cement supply so much larger than for steel?
- ii) Why is the uncertainty for cement demand in 2050 so much smaller than for steel?
- iii) Is it possible to say something about what the most limiting factors are for the supply of steel versus cement: electricity supply, CCS capacity or the carbon budget?
- iv) It is surprising that the curves for both cement and steel first go down and then around 2040 or so exhibit an increasing trend again. How can this be explained?

>Response

Thank you for this helpful suggestion. We have added the following explanation in the main text to clarify these points:

“A comparison of steel and cement highlights three caveats. First, the uncertainty in cement supply is greater than the uncertainty in steel supply. This is due to the relatively limited options for decarbonizing the cement production process, which depends exclusively on the deployment of CCUS. Second, the range of expected demand is greater for steel than for cement. This discrepancy likely stems from the limited exploration of demand reduction strategies for cement. In fact, the LED scenario ²⁵ only considers one strategy for cement due to the lack of scientific evidence, while a broader range of strategies are examined for steel. Third, the feasible supply of both steel and cement exhibits an initial downward trend, followed by an upward trend. This pattern is attributed to the combined effects of the shape of the carbon budget, which requires immediate emission reductions, and the linear deployment of zero-emission infrastructure, which is assumed to increase steadily over time.”

Comment

Section “Certain growth of recycling in an uncertain future”

32. This section is only about steel, which is not obvious from the heading. The heading could be rephrased to communicate this.

>Response

We agree with the reviewer and have added “steel” to the heading.

Comment

33. For steel, CCS is usually applied to fossil-based production technologies, such as BF-BOF or Fossil DRI-EAF. However, their production share is very low in 2050, while the CO₂ captured seems to potentially further increase according to Fig. 1c). Thus, the question arises whether the results for carbon captured from steel production in 2050 practically make sense, or would they imply that CCS is also applied to H₂-DRI or scrap-EAF?

>Response

Thank you for your question. You are absolutely right, CCUS is typically applied to fossil-based production. In our calculations, CCUS is applied mostly to fossil-based production technologies; the upper bound for fossil-based production emits almost the same amount of CO₂ as the upper bound for CCUS implementation.

Comment

Figure 3:

34. Fig. 3b): the trajectory of fossil DRI-EAF is not explained. It seems that it relates to figure S3 in the supplementary information. Why does it not show higher uncertainties?

47. Equation 4: Cap_FDRI is not explained. Does it relate to figure S3 in the supplement?

61. This figure is not mentioned or explained in the manuscript. What is the reasoning behind the choice of the values illustrated in the figure? Is the trajectory not subject to uncertainty?

>Response

Thank you for bringing these points to our attention. We have added the following explanations to clarify the points.

*“Fossil fuel-based DRI production, while having a better emission profile than blast furnace-based production, is highly dependent on regional fossil fuel availability²⁶. Consequently, its production levels are expected to experience moderate changes (**Supplementary Fig. 4**).”*

*“**Supplementary Fig. 4** Maximum deployment of fossil DRI-EAF route. Fossil fuel-based DRI production is strongly influenced by regional factors and has been predominantly deployed in the Middle East and India, where significant fossil fuel reserves exist²⁶. In this study, the maximum deployment of fossil fuel-based DRI production is determined exogenously based on a detailed analysis conducted by the IEA⁷. The IEA’s analysis takes into account various factors such as regional fossil fuel availability, technological feasibility, and environmental considerations to establish the upper limit for the deployment of this production technology.”*

Comment

Section “Sufficient feasible supply to meet basic human needs”

35. L. 190: what about uncertainty in this data from reference 27? What are the assumptions regarding the development of the world population? Do these not vary depending on scenarios?

>Response

Thank you for raising this important point. We have included the uncertainty of the minimum material requirements to meet basic human needs. Our study uses the same future population data as the literature to ensure consistency.

Comment

36. L. 400 – 404: this seems to be an important information. It could be valuable to move it out of the figure caption into the main text.

>Response

Thank you for pointing this out. We have revised the text as follows to emphasize this point in the main text. To keep the text flowing, we have left the caption descriptions intact.

“The challenge lies in the substantial material demand generated by high-income countries for stock replacement and expansion, with houses and cars becoming larger and heavier year after year²⁸.”

Comment

37. L. 402: What is the definition of the “required level”?

>Response

Thank you for this question. We have revised the text in the figure caption accordingly.

Comment

38. L. 400 – 404: What does this fact imply? It seems to influence the interpretation of the results, especially for cement where recycling is not common practice. Does this mean that the difference between the feasible supply and the min. requirements for basic human needs could be interpreted smaller than shown in fig. 4?

>Response

Thank you for this question. Yes, you are right. The challenge is that if we do not reduce material demand in high-income countries, we will not be able to provide sufficient material supplies to those who really need them to meet basic human needs. We have revised the text as described above to highlight this important point:

Comment

Discussion:

39. L. 222-224: what about the challenge of exceeding the carbon budget? If industry does not receive the required infrastructure for low-carbon production technologies, they might continue using the current production technologies to satisfy the demand, which will lead to higher emissions than allowed by the carbon budget.

>Response

Thank you for this insightful comment. This point was certainly not sufficiently emphasized in the earlier manuscript. We have added the following discussion to clarify the point:

*“If these challenges are not addressed head-on, the gap between feasible supply and expected demand risks being filled by emission-intensive production processes. This could result in cumulative emissions of ~160 Gt-CO₂ by 2050, representing ~40% of the remaining 1.5°C budget or ~20% of the remaining well-below 2°C budget (**Supplementary Fig. 6**).”*

Comment

40. L. 237 – 238: It seems to me that this sentence belongs to the next paragraph which describes the demand reduction strategies.

>Response

Thank you for pointing this out. We have revised the text accordingly.

Comment

41. L. 240 – 255: The strategies described in this paragraph are great suggestions. Currently, it is phrased in a neutral way such that it is not clear how these strategies could be best realized or incentivized. To which extent is it the responsibility of governments to define respective policies or regulations? If it is their responsibilities, this could be added as a challenge in line 223.

43. The results of this manuscript are worrying as they suggest that there might be a shortage of steel and cement supply in the future under the 1.5° carbon budget. Based on the results and possible measures presented in the discussion, is it possible to say something about which measures might be the most effective to minimize the supply-demand-gap?

>Response

Thank you for these critical comments. We have intentionally kept this part of the description neutral, because the aim here is to show various potential for improving resource efficiency, not to specify the most effective way. We agree with the reviewer that it would be helpful to clarify the most effective way and the role of government. However, such an argument cannot be directly supported by our analysis and requires a more detailed policy analysis. Therefore, we have limited ourselves to asserting the importance of balancing the discussion between supply-side and demand-side actions, which can be well supported by our analysis.

Comment

42. L. 288: How can bulk materials be part of the solution to the ongoing climate crisis? This conclusion seems to be not supported by the results presented in the paper.

>Response

We thank the reviewer for highlighting this point. We have removed this text to avoid this overstatement.

Comment

Methods:

Model overview:

44. L 297: please specify the “zero-emissions infrastructure”

>Response

Thank you for this comment. We have specified this point in the text.

Comment

45. Equation 2: $CI_i(t)$ seems to be not defined. Please add a definition in form of an equation. Does this include also the emission intensity of electricity from fig. 2f)?

>Response

Thank you for bringing this point to our attention. We have revised the equation (2) accordingly.

Comment

46. It would be very helpful to the reader to add references for the data presented in figure 2 to the respective variables in this section.

>Response

Thank you for this helpful suggestion. We have listed the main literature sources in the methods section and added the summary table in the supplementary information.

“The main data sources are based on various academic papers and reports: system variables and parameters governing physical mass balance ^{47–50}; emission profiles of material production ^{51–57}; energy penalty of carbon capture ^{58,59}; and future population ⁶⁰. More details can be found in the supplementary information.”

Comment

48. Equations 6-8: It would be valuable to add a flowchart of the system explaining the definition of the different variables used in these equations. This could go in the supplementary information.

>Response

Thank you for this suggestion. We have created the diagram of the system and added it to the supplementary information. We hope this diagram will help readers better understand our model.

Supplementary Fig. 2 Conceptual flow chart of iron and steel flows. The symbols at the bottom right of the process show the associated coefficients; the three equations show scrap estimation formulas based on simple mass balance.

Comment

49. L. 319: Please correct the grammar of this sentence.

>Response

Thank you for pointing this out. We have corrected the grammar accordingly.

Comment

50. L. 326-327: Please refer to the variables here as defined in the equations. “Carbon capture capacity” does not fit to the variable names mentioned for CCS.

>Response

Thank you for pointing this out. We have revised the text accordingly.

Comment

51. The methods currently barely name any data sources or quantitative assumptions. Some are mentioned in the main part of the manuscript, some in the supplementary information, some on the github repository. Please explicitly state data sources and assumptions in the methods section or refer to the respective alternative position where they may be specified.

>Response

We thank the reviewer for highlighting this important point. We have modified our explanation in this domain as discussed above.

Comment

Carbon budgets:

52. L. 336: What are the annual emission mitigation rates. Could they be specified more in detail or a reference added?

>Response

Thank you for this question. We hope the following revised text gives a bit clearer explanation: “We allocate the total carbon budgets to the global steel and cement sectors by multiplying the current emissions of the steel and cement sectors by the annual emissions reduction rate ⁶¹ (**Supplementary Fig. 3**).”

Comment

Supplementary information:

Table S1:

53. What is the role of the “advanced BF-BOF”, it is not mentioned in the main part of the manuscript.

55. The electricity consumption of H2-DRI-EAF can largely vary depending on the assumptions of the technology (Bhaskar et al. 2020, Figure 12). What are your technology assumptions, e.g. efficiency of the electrolyzer?

56. How are the target values used in the model? Do the 4 parameters change over time? If so, how? This is not mentioned in the manuscript. Why are such individual improvements assumed for cement, but not for steel, where e.g. efficiency improvements could also apply?

>Response

We thank the reviewer for this thought-provoking comment. We have added the following section to clarify these points:

“Technological progress

This study assumes that technological progress in the industry will align with established roadmaps and industry-accepted best practices. Specifically, we envision top gas recycling and coke substitution in blast furnaces and improved post-consumer scrap recovery for the steel sector. For the cement sector, our assumption includes improved energy efficiency, clinker

substitution, and fuel substitution. Detailed information on the level of improvement can be found in the supplementary information.

In this domain, four key assumptions deserve attention. First, the hydrogen-based direct reduction system is based on the literature⁵⁵ with an electrolyzer efficiency of 45 kWh/kg H₂ and a hydrogen mass flow rate of 1.5 (i.e., 50% oversupply of hydrogen for full conversion of iron ore in the shaft⁸). Since this electrolyzer efficiency is already at a high level⁶⁴, no further efficiency improvement is assumed. Second, clinker has traditionally been substituted mainly as fly ash and granulated blast furnace slag (GBFS), which will decline in a decarbonized future⁶⁵. Therefore, clinker substitution will have to be provided as resources other than fly ash and GBFS. Our assumption here is based on recent studies that show significant potential for substitution with calcined clay, agricultural by-product ash, forestry by-product ash, and end-of-life binders^{66,67}. Third, similar to supplementary cementitious materials, the cement industry currently uses waste materials as thermal fuels, which may become less available in a decarbonized, more circular future⁵³. We assume an expanded use of waste from agricultural, chemical, and food production⁵⁷. It is important to note that a more comprehensive assessment of resource availability needs to consider the interconnected system beyond just the steel and cement industries⁶⁸. Fourth, technological improvements are assumed to be achieved linearly over the period 2050. This assumption, although simplistic, is based on the limited evidence and is consistent with industry roadmaps that occasionally assume linear progress⁵⁷."

Comment

54. I suppose that the electricity consumption includes the electricity for hydrogen electrolysis. If so, please add this information.

>Response

You are right. We have added the explanation accordingly. Thank you for pointing this out.

Comment

Table S3:

57. This table is neither referred to in the manuscript, nor is it explained how its data is used in the model.

58. Please add an explanation for how the model incorporates energy penalty for CCS.

59. Which energy carriers and resulting emissions are assumed for the energy requirement for CCS?

60. Why is the energy penalty for steel CCS so low. Which assumptions are taken for this technology? Depending on the choice of CCS technology, the current assumption seems too low (Perpinan et al. 2023, figure 16).

>Response

Thank you very much for emphasizing these important points. We have revised the equations and added the following table.

Supplementary Table 4 Overview of carbon capture technologies and energy penalties. Data are based on Refs^{24,25}.

Overview	Value
Steel sector This study assumes the use of post-combustion with membranes due to the high technology readiness level. The system can be effectively operated using electrical energy.	1.7 GJ/t-CO ₂
Cement sector This study assumes the use of membrane-assisted CO ₂ liquefaction, which has the advantage of being easily retrofitted and requires only electricity as an input to the process.	3.2 GJ/t-CO ₂

REVIEWER COMMENTS

Reviewer #2 (Remarks to the Author):

Response to review round 1: My comments during Review 1 have been answered in detail. Thank you for your extensive consideration of the suggestions and your clear reasoning why not to follow certain suggestions, when this was the case.

New comments:

General:

1) I welcome the decision of the authors to additionally consider the well-below 2C scenario next to the 1.5C scenario. It adds more detail and differentiation to the findings – well done! However, I miss some more referral to the results of the well-below 2C scenario (apart from the feasible supply figure, they are only shown in the SI). It would be great to see a bit more reflection of the 2C scenario results also in the main body (l. 200 to 250, linking to figure SI 7 & 8).

2) Supply demand gap (SI fig 6): could it also be shown for well-below 2C?

3) At multiple sections, the authors emphasize that their work also points to the “level of resource efficiency required under the feasible supply” (e.g. l.307). Could the authors elaborate more on the actual values for required resource efficiency based on their model? Unfortunately, I could not find it clearly in the text. However, the way I interpret the abstract (“Our feasible supply scenarios [...] establish benchmarks for the required level of resource efficiency”), one expects to find specific values in the paper. If you will not provide actual benchmarks, you could consider softening the phrasing.

By line:

l. 49 “are” -> “is”

l. 146-149: I think it could be of value for the reader to explicitly state the total volume (Gt) of the carbon budgets, either here or in the Methods sections.

l. 155: “Paris-compliment” -> “Paris-compliant”, maybe a typo?

l. 386-388: It is not clear to me what values the annual emission reduction rates refer to, for both materials, particularly since the additional information in the Supplementary Fig 3 description was removed. The cited source 61 (Raupach et al, 2014) offers three allocation types. Could you elaborate in more detail how the carbon budgets are constructed for the sectors?

l. 408: substituted mainly as -> substituted mainly by

l. 410: be provided as -> be provided by

Overall, I recommend the publication of this work, if general comment 1 & 3 are implemented (or, alternatively, sound arguments are provided why the authors disagree with these suggestions). The article provides additional, meaningful insights into the feasible future material supply from emission-intensive steel & cement. The novel approach of ‘forecasting supply and backcasting demand’ is valuable and offers a check of the usual scenario modelling approaches. The manuscript has been revised thoroughly, is well written and results are presented in plots of high quality.

Reviewer #3 (Remarks to the Author):

Most of the previous comments have been well addressed by the authors. If the comments below can be resolved, I would recommend the article for publication.

Major comments:

1. Follow-up question and comment to supplementary table 4 (former table 3): It is great to see that

you adjusted the details about the CCS technologies and increased the energy penalties. However, some questions are still open:

i) The technology choice for the post-combustion membrane technology is somewhat surprising as this technology is rather novel, not very well studied and not very common in other scenario studies yet. Thus, the assumptions may be of higher uncertainty than for other technologies, such as post-combustion with chemical absorption using monoethanolamine. If this was a deliberate choice of the authors, a note to the reader about the novelty of the technology could be very helpful to communicate potential implications and uncertainties.

ii) I am surprised that for steel the chosen energy penalty of 1.7 GJ/t-CO₂ is lower than the average of 1.8 GJ/t-CO₂ which is reported in the mentioned reference of Perpinan et al. 2023. What is the reasoning behind this assumption?

iii) Even 1.8 GJ/t-CO₂ seems quite low and optimistic to me. I invite the authors to reconsider and double-check whether they are confident with this value in combination with the assumptions about the t CO₂ captured per t steel produced.

iv) What is the assumption for t CO₂ captured per t steel or t cement produced? This assumption could also be added to the table.

Minor points:

2. Figure 1c) is for me still not very easy to understand or to reproduce for the following reasons:

i) Fig. 1.c) Max. value in 2050 for steel: this is set to 670 Mt-CO₂ based on 1 IEA scenario, but in figure 1.b) the max. value for steel in 2050 is ca. 1000 Mt-CO₂ for the scenario IEA (2010) (first bar in fig. 1.b). I assume that the max. values are chosen based on the max. combined CO₂ captured which is probably the very right bar in Fig. 1.b) to consider the systemic approach of the IEA scenarios (instead of selecting the max. value of steel and cement separately independent of the scenarios). This would make sense however it is not completely clear from the text and could be explained in a few words.

ii) Fig. 1.c): The idea of the overlapping areas is not easy to perceive in the figure, especially since "steel" is written in the overlap area, which suggests that this area is only for steel. I would recommend adding coloured lines to the borders of each area (i.e. the min. and max. line of the cement and steel area) and adding a legend again showing cement and steel where these borders are also visible. Thereby, the labels within the overlap area could be eliminated which might help.

iii) Fig. 1c) 2 bars on the right: it seems uncommon to me to show a range of values with the current choice of "normal" bars. It could help to use error bars instead as it has also been done in other figures in this manuscript to better convey the meaning of the bars.

3. Supplementary tables are not referred to in the article, apart from table 5 and 6. It would be great to add the cross-references in the article to guide the reader to the respective section where this data was used in.

Response to reviewers

Comment

>Response

Thank you again for taking your valuable time to read our manuscript. We are pleased to hear that your concerns during Review 1 have been adequately addressed.

Comment

of the authors to additionally consider the well-below 2C scenario next to the 1.5C scenario. It adds more detail and differentiation to the findings – well done! However, I miss some more

>Response

This is indeed a great idea. We have added the following text and figures to better reflect the well-below 2 °C budget:

*“These trends remain the same with the well-below 2 °C budget; the difference between the 1.5 °C budget and the well-below 2 °C budget is mainly in the rate of phase-out of blast furnace-based production, with the well-below 2 °C budget showing a more gradual phase-out by 2050 (**Supplementary Fig. 7**).”* (Lines 213-216)

“The minimum requirements represent an even smaller fraction of the feasible supply under the well-below 2 °C budget: 11-12% (interquartile range) for steel and 39-45% (interquartile range) for cement. Theoretically, therefore, the basic needs of the growing world population could well be met by the feasible supply, even if the zero-emissions infrastructure is deployed at the lower end of the potential range.” (Lines 236-241)

*“As shown in **Fig. 5**, current per capita material stocks in high-income countries far exceed the feasible stock levels derived from the 1.5-2 °C budgets, while low-income countries have material stocks well below the feasible levels.”* (Lines 244-247)

Fig. 4 Cumulative feasible supply of steel and cement compared to the minimum requirements to meet basic human needs, 2015-2050.

Fig. 5 Feasible per capita in-use stocks of steel and cement in 2050 compared to the current levels across four income groups.

Comment

>Response

Thank you for highlighting this point. We have added the figure for the supply-demand gap under the well-below 2 °C budget.

Supplementary Fig. 9 Gap between feasible supply and expected baseline demand. The solid black line shows the median, and the colored bands around it show the interquartile range. We assume that the gap between feasible supply and expected demand for steel and cement will be filled by the current emission-intensive production processes (i.e., BF-BOF). This would entail carbon emissions of approximately 2.1 t-CO₂/t-steel or 0.6 t-CO₂/t-cement. The consequences of such emissions would be severe, potentially resulting in cumulative emissions of up to ~160 Gt-CO₂ by 2050. This amount represents ~40% of the remaining 1.5°C budget or ~20% of the remaining well-below 2°C budget.

Comment

3) At multiple sections, the authors emphasize that their work also points to the “level of resource efficiency required under the feasible supply” (e.g. I.307). Could the authors elaborate more on the actual values for required resource efficiency based on their model? Unfortunately, I could not find it clearly in the text. However, the way I interpret the abstract (“Our feasible supply scenarios [...] establish benchmarks for the required level of resource efficiency”), one expects to find

>Response

We thank the reviewer for pointing this out. We have revised the figure and text as follows to clarify this point:

“The approach is also advantageous because it indicates the specific level of resource efficiency required under the feasible supply: we will need to provide the same level of services with 60% less material use in construction and 40% less in manufacturing to stay within the 1.5 °C budget.” (Lines 312-316)

Comment

I. 49 “are” -> “is”

>Response

Modified.

Comment

>Response

We have revised the text as follows:

“The carbon budget is based on limiting the global mean temperature rise within 1.5 °C with a 50% probability, consistent with the Paris Agreement pledges (~420 Gt-CO₂)³. We also consider a carbon budget for a 50% probability of 1.7°C (equivalent to an 83% probability of 2.0°C), which corresponds to “well below 2°C” (~770 Gt-CO₂)²⁴.” (Lines 392-395)

Comment

I. 155: “Paris-compliment” -> “Paris-compliant”, maybe a typo?

>Response

Modified.

Comment

>Response

We apologize for our unclear explanation in the previous manuscript. We have added and revised the following figures to clarify this point:

Supplementary Fig. 3 Annual emissions mitigation rate for Paris-compliant carbon budgets. The carbon budget is based on limiting the global mean temperature rise within 1.5 °C with a 50% probability, consistent with the Paris Agreement pledges (~420 Gt-CO₂)²⁶. We also consider a carbon budget for a 50% probability of 1.7°C (equivalent to an 83% probability of 2.0°C), which corresponds to “well below 2°C” (~770 Gt-CO₂)²⁷. The annual emissions mitigation rate is determined using equation 3 in Ref²⁸.

Supplementary Fig. 4 Paris-compliant carbon budgets for the steel and cement sectors. We allocate the total carbon budgets to the global steel and cement sectors by multiplying the current emissions of the steel and cement sectors by the annual emissions mitigation rate shown in **Supplementary Fig. 3**.

Comment

>Response
Modified.

Comment

>Response

Modified.

Comment

cement. The novel approach of 'forecasting supply and backcasting demand' is valuable and

>Response

Thank you for your positive feedback! Your comments during reviews 1 and 2 have greatly improved this manuscript.

Comment

>Response

Thank you again for taking your valuable time to read our manuscript. We are pleased to hear that most of your concerns during Review 1 have been adequately addressed.

Comment

>Response

We thank the reviewer for highlighting this critical point. We have revised and added the following text to clarify this point:

Supplementary Table 4 Overview of carbon capture technologies and energy penalties. Data are based on Refs ^{24,25}. Note that our simplified model does not take into account detailed processes within the plant (e.g., CO₂ purity and capture rates). Therefore, the model only considers the potential range of total carbon capture in the sector and the energy penalty to estimate the maximum global materials production within Paris-compliant carbon budgets.

	Overview	Value
Steel sector	This study assumes the use of post-combustion with membranes due to the high technology readiness level ²⁴ (*). The system can be effectively operated using electrical energy.	1.8 GJ/t-CO ₂
Cement sector	This study assumes the use of membrane-assisted CO₂ liquefaction, which has the advantage of being easily retrofitted and requires only electricity as an input to the process ²⁵.	3.2 GJ/t-CO ₂

(*) It should be noted that many previous studies have assumed post-combustion with chemical absorption as the carbon capture technology in the steel sector ²⁴. Therefore, there is significant uncertainty in the energy penalty of post-combustion with membranes, which can be up to 4.4 GJ/CO₂. We explore the impact of this uncertainty in the form of a sensitivity analysis (**Supplementary Fig. 8**).

Comment

>Response

Thank you very much for pointing this out. We derived an average of 1.7 GJ/CO₂ from Table 8 in Perpnan et al. 2023, which should be 1.8 GJ/CO₂ according to the paper. We believe this difference is due to rounding of the data in Table 8, so we now use 1.8 GJ/CO₂.

Comment

>Response

Thank you for this comment. We have added the sensitivity analysis for the high energy penalty case:

Supplementary Fig. 8 Sensitivity of feasible steel supply within a 1.5 °C budget to the energy penalty of carbon capture technologies. (a) Base case: 1.8 GJ/t-CO₂. (a) High energy penalty case: 4.4 GJ/t-CO₂. The high energy penalty case has a wider range of feasible supply uncertainty and a median in 2050 that is about 7% lower than the base case.

Comment

>Response

Thank you for this comment. We have added the following text to clarify this point:

“Note that our simplified model does not take into account detailed processes within the plant (e.g., CO₂ purity and capture rates). Therefore, the model only considers the potential range of total carbon capture in the sector and the energy penalty to estimate the maximum global materials production within Paris-compliant carbon budgets.” (SI)

Comment

>Response

Thank you for this comment. We have revised the text as follows:

“First, the upper bound of CCUS deployment is based on the IEA Net Zero scenario ¹⁹, which considers the most ambitious CCUS capacity in the steel and cement sectors combined.” (Lines 110-112)

Comment

since “steel” is written in the overlap area, which suggests that this area is only for steel. I would

iii) Fig. 1c) 2 bars on the right: it seems uncommon to me to show a range of values with the current choice of “normal” bars. It could help to use error bars instead as it has also been done in

>Response

Thank you for these great suggestions. We have revised the figure accordingly:

Fig. 1 Potential range of future global zero-emissions infrastructure deployment. (a) Carbon capture capacity for 2021 projected by the IEA scenarios. (b) Carbon capture capacity for 2050 projected by the IEA scenarios. (c) The potential range of future carbon capture capacity. (d) Total electricity supply projected by the IEA scenarios. (e) The potential range of future total electricity supply. (f) The potential range of future emission intensity of electricity. The data are based on a series of Energy Technology Perspectives reports ¹⁷. We examined all reports and extracted data from those reports for which data were available. Current operating and planned carbon capture capacities for 2030 were obtained by accessing the IEA database in June 2023 ¹⁸. The right-hand error bars in Fig. 1c show the range of 2050 values.

Comment

>Response

This is indeed a great idea. We have added the cross-references in the main text.

REVIEWER COMMENTS

Reviewer #2 (Remarks to the Author):

The authors have sufficiently addressed the feedback given in this second round of review and the quality of the analysis has been further improved through these refinements.

While reading the final documents, I have found following final suggestions for improvement:

Fig S8: (a) High energy penalty case: 4.4 GJ/t-CO₂ ->(b)

Fig S1, S4 and S9: you could consider adding letters to subplots in these plots, too.

I congratulate the authors to this very interesting and valuable contribution to the feasibility of future provision of steel and cement and recommend the article for publication.

Reviewer #3 (Remarks to the Author):

The authors addressed all comments and further improved the manuscript. The last additions are very helpful, such as the additional figures about the carbon budgets or the sensitivity analysis of the energy penalty of carbon capture technologies. They increase the overall clarity, transparency, reproducibility as well as the scientific soundness of the work. It is great to see that the authors decided to add these additional details.

There is one comment which is still not clear to me:

Lines 110-112, for figure 1.c):

Thank you for adding the text: "First, the upper bound of CCUS deployment is based on the IEA Net Zero scenario 19, which considers the most ambitious CCUS capacity in the steel and cement sectors combined."

The rationale seems reasonable to me. However, the reference to the IEA Net Zero scenario appears for the very first time here, if I am not mistaken. It is not mentioned in the figure caption or figures 1.a) or 1.b). The numbers specified seem to match with the bars of IEA (2023) in figure 1.b), but the references are not the same (IEA 2023 is a different reference than IEA Net Zero). Thus, it is not clear how the upper bound of fig. 1.c) can be derived from fig. 1.a) or b). This could be resolved, e.g. by either aligning the references or adding the data of the IEA Net Zero scenario in figures 1.a) or b), in case that the data is indeed still missing there. Alternatively, some explanation how the scenarios, references and scenario labels relate to each other could be helpful.

Furthermore, I have a few very minor comments regarding notation:

- 1) Lines 356-366: In the explanations of the variables, the dependency on time is not mentioned in the notation, i.e. (t). It would be great to add the "(t)" respectively to make clear which of the variables are time-dependent and which ones are constants.
- 2) Line 379: CCUS(t) is not a variable in the preceding equations, should this be CC(t) instead?
- 3) Figure 8 in the supplement: figure caption has a typo, it mentions (a) twice instead of (a) and (b)

Overall, I recommend the article for publication. The analysis is very valuable and provides new insights. It is well and clearly presented. The comments which are still open are not related to the content but only to the notation.

Response to reviewers

Reviewer #2 (Remarks to the Author):

Comment

The authors have sufficiently addressed the feedback given in this second round of review and the quality of the analysis has been further improved through these refinements.

>Response

We would like to express our sincere gratitude for your diligent review of our manuscript throughout the review process. Your constructive feedback and insightful suggestions have played a pivotal role in refining the quality of our analysis, and we are pleased to hear that these refinements have been effective.

Comment

While reading the final documents, I have found following final suggestions for improvement:
Fig S8: (a) High energy penalty case: 4.4 GJ/t-CO₂ ->(b)

>Response

Thank you for pointing out our error. We have corrected the text accordingly.

Comment

Fig S1, S4 and S9: you could consider adding letters to subplots in these plots, too.

>Response

Thank you for this suggestion. We have added letters to subplots.

Comment

I congratulate the authors to this very interesting and valuable contribution to the feasibility of future provision of steel and cement and recommend the article for publication.

>Response

Thank you very much!

Reviewer #3 (Remarks to the Author):

Comment

The authors addressed all comments and further improved the manuscript. The last additions are very helpful, such as the additional figures about the carbon budgets or the sensitivity analysis of the energy penalty of carbon capture technologies. They increase the overall clarity, transparency, reproducibility as well as the scientific soundness of the work. It is great to see that the authors decided to add these additional details.

>Response

We sincerely appreciate your detailed and constructive feedback throughout the review process. Your comments have been instrumental in shaping our manuscript, and we are pleased to hear that the additions have helped to improve the overall quality of our work.

Comment

There is one comment which is still not clear to me:

Lines 110-112, for figure 1.c):

Thank you for adding the text: "First, the upper bound of CCUS deployment is based on the IEA Net Zero scenario 19, which considers the most ambitious CCUS capacity in the steel and cement sectors combined." The rationale seems reasonable to me. However, the reference to the IEA Net Zero scenario appears for the very first time here, if I am not mistaken. It is not mentioned in the figure caption or figures 1.a) or 1.b). The numbers specified seem to match with the bars of IEA (2023) in figure 1.b), but the references are not the same (IEA 2023 is a different reference than IEA Net Zero). Thus, it is not clear how the upper bound of fig. 1.c) can be derived from fig. 1.a) or b). This could be resolved, e.g. by either aligning the references or adding the data of the IEA Net Zero scenario in figures 1.a) or b), in case that the data is indeed still missing there. Alternatively, some explanation how the scenarios, references and scenario labels relate to each other could be helpful.

>Response

Thank you very much for pointing out our error. Indeed, the text should have referred to IEA (2023). We have revised the text and reference accordingly.

"First, the upper bound of CCUS deployment is based on the 2023 IEA report ¹⁹, which considers the most ambitious CCUS capacity in the steel and cement sectors combined." (Lines 110-112)

Comment

Furthermore, I have a few very minor comments regarding notation:

1) Lines 356-366: In the explanations of the variables, the dependency on time is not mentioned in the notation, i.e. (t). It would be great to add the "(t)" respectively to make clear which of the variables are time-dependent and which ones are constants.

>Response

Thank you for this suggestion. We have revised the text to clarify which variables are time-dependent.

Comment

2) Line 379: CCUS(t) is not a variable in the preceding equations, should this be CC(t) instead?

>Response

Indeed, many thanks for pointing this out. We have revised it to be CC(t).

Comment

3) Figure 8 in the supplement: figure caption has a typo, it mentions (a) twice instead of (a) and (b)

>Response

Fixed. Thank you for pointing this out.

Comment

Overall, I recommend the article for publication. The analysis is very valuable and provides new insights. It is well and clearly presented. The comments which are still open are not related to the content but only to the notation.

>Response

We sincerely appreciate your recommendation for publication and your positive feedback on the value and presentation of our analysis.